# PRISM: Learning Realistic Depth via Physics-Grounded Noise Disentanglement with Semantic-Geometric Collaboration

**Xiujian Liang** [1 2]  **Jiacheng Liu** [2 3]  **Mingyang Sun** [2 3]  **Qichen He** [2 4]  **Anda Cheng** [5]  **Cewu Lu** [4 2]  **Jianhua Sun** [4 2]

## Abstract

Real-world physical sensing exhibits complex, heterogeneous noise patterns that deviate significantly from idealized simulation, posing a fundamental bottleneck for sim-to-real transfer. Existing sensor modelings typically treat depth noise as a monolithic black-box process, overlooking the distinct physical mechanisms that govern different error modalities. In this work, we introduce a physics-grounded paradigm that disentangles monolithic noise into two complementary modalities: **sensing invalidation** and **measurement inaccuracy**, enabling a tailored treatment of noise sources based on their physical origins. Building on this insight, we propose **PRISM**, a tripartite framework that distills 3D Visual Foundation Model features as rich spatial-semantic priors for physics-based reasoning. To address the inherent sparsity and class imbalance of invalidation regions, we develop Hierarchical Positive-Prioritized Supervision, integrating multi-scale positive-weighted objectives with a positive-preserving dynamic hard mining strategy to enforce precise artifact delineation. Extensive benchmarks demonstrate that PRISM achieves state-of-the-art fidelity in noisy depth synthesis. Furthermore, downstream robotic experiments show that PRISM facilitates a **92.1%** average success rate in the real world, marking a significant improvement over monolithic baselines.

## 1. Introduction

Simulation-based training has emerged as a cornerstone of robotic learning, offering unparalleled advantages in safety, scalability, and data availability, enabling the ex-

[1]Fudan University [2]Shanghai Inovation Institute [3]Zhejiang University [4]Shanghai JiaoTong University [5]Ant Group. Correspondence to: Jianhua Sun <gothic@sjtu.edu.cn>.

*Proceedings of the 43rd International Conference on Machine Learning*, Seoul, South Korea. PMLR 306, 2026. Copyright 2026 by the author(s).

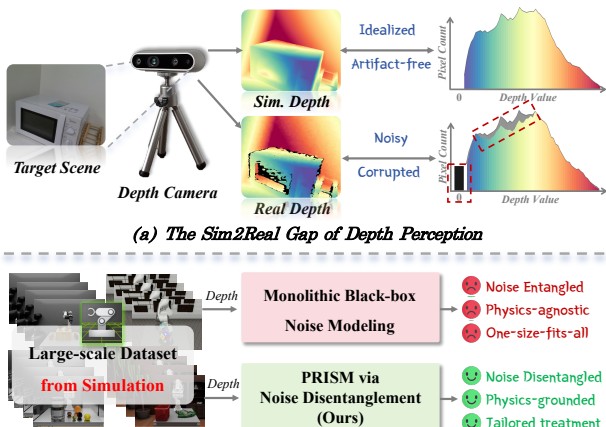

*(a) The Sim2Real Gap of Depth Perception*

*(b) The Comparison of Mainstream Approach with Our PRISM*

*Figure 1.* **The Anatomy of Depth Perception and Modeling.** (a) The Reality Gap: Unlike pristine simulation, real-world physical sensing exhibits a bimodal noise distribution: black voids and gray residuals. (b) Paradigm Shift: While prior methods conflate noises into a monolithic modeling, our PRISM framework disentangles noise into distinct modalities based on physics-grounded structural priors, enabling targeted synthesis for each noise source.

ploration of complex manipulation and navigation tasks at scale(Mittal et al., 2023). However, the deployment of simulation-trained policies remains fundamentally bottlenecked by the *sim-to-real gap*(Jia et al., 2025). Among all sensing modalities, depth perception stands out as particularly susceptible: while simulated depth is geometrically pristine, consumer-grade sensors exhibit complex, heterogeneous noise patterns(Fig.1a). Unlike the visual gap in the RGB, the depth gap introduces functional distortions that directly corrupt geometric reasoning, misleading downstream perception and control(Wang et al., 2022).

To bridge this geometric gap, prior efforts have bifurcated into two primary paradigms. **Explicit analytical modeling** relies on predefined heuristics(Borrego et al., 2018) or differentiable ray-tracing (Zhang et al., 2023b) to simulate light transport. While theoretically rigorous, these methods often suffer from limited generalization beyond specific sensor models or rigid calibration requirements. Conversely, **implicit data-driven modeling** employs image translation (Shen et al., 2022) or diffusion models (Zhou et al., 2025; Xu et al., 2025) to learn noise distributions from

data. Crucially, despite their methodological divergence, both paradigms typically model depth noise as a **monolithic stochastic process**. By indiscriminately conflating distinct physical mechanisms into a unified representation, they fail to reproduce the spatially correlated artifacts inherent to physical sensing, often hallucinating noise in clean regions or failing to predict structural voids.

We argue that effective noise modeling should be grounded in the physical mechanisms of sensor degradation, which are intrinsically linked to scene semantics. Real-world depth corruption follows structured physical regularities: specular surfaces (e.g., metal, mirror) cause regular reflection resulting in **Sensing Invalidation**, whereas transparent or translucent materials (e.g., glass, plastic) induce refraction or sub-surface scattering, leading to **Measurement Inaccuracy**. Unlike monolithic models that indiscriminately conflate these mechanisms, PRISM establishes a **physics-grounded noise disentanglement** paradigm. We posit that this bimodal decomposition is a pragmatic macroscopic taxonomy supported by sensor physics, necessitating specialized components to model their statistical and physical properties of each error source.

To operationalize this insight, we propose **PRISM** (**P**hysics-**R**easoned **I**mplicit **S**ensor **M**odeling), a semantic-geometric collaborative framework designed to ′refract′ monolithic sensor noise into physically motivated modalities. The framework is instantiated as a tripartite synergistic architecture following a structured pipeline: *Reason → Disentangle → Generate*. First, the **Semantic-Physics Reasoner** (SPR) leverages a frozen 3D VFM (e.g., DepthAnything(Lin et al., 2025), MoGe2(Wang et al., 2025b)) to extract semantic-aware features that act as physics priors identifying non-Lambertian surfaces. These priors are then injected into the **Bimodal Noise Disentangler** (BND), which fuses semantic context with simulation geometry to implicitly predict the pixel-wise probability of invalidation and transform into a region mask. Crucially, instead of predicting noise directly, BND formulates disentangled representations that condition a **Noise Residual Generator** (NRG). NRG employs a diffusion-based process, inspired by (Xu et al., 2025), to synthesize high-fidelity residual maps, ensuring generated artifacts are not only statistically plausible but also geometrically consistent with the underlying scene structure.

Furthermore, training this framework presents a unique optimization challenge: invalidation regions are spatially sparse (often $< 10\%$ of pixels). Standard losses inevitably collapse under this extreme class imbalance underweighting these rare but critical events. To counter this, we develop **H-PPS** (**H**ierarchical **P**ositive-**P**reserving **S**upervision). Unlike conventional hard example mining discards rare signals, H-PPS combines multi-scale boundary constraints with recall-prioritized mining protocol.

Our contributions are summarized as follows:

**1) Physics-Grounded Disentanglement:** We fundamentally reframe depth noise modeling from a monolithic stochastic process to a physics-grounded taxonomy. We identify *Sensing Invalidation* and *Measurement Inaccuracy* as two physically motivated modalities, providing a theoretical basis for treating signal loss and signal bias separately.
**2) Semantic-Geometric Collaboration:** We propose PRISM, a unified framework that distills the rich physical common sense of 3D Visual Foundation Model to drive noise synthesis. By explicitly aligning semantic priors with geometric distortion, we achieve noise generation that is not only visually realistic but physically consistent with scene.
**3) Sparsity-Aware Optimization:** We address the sparsity and imbalance in modeling sensing invalidation through Hierarchical Positive-Prioritized Supervision. By integrating multi-scale positive weighting with a positive-preserving dynamic hard mining strategy, we solve the optimization dilemma of learning sparse, boundary-sensitive artifacts.
**4) SOTA Fidelity and Sim2Real Transfer:** Extensive benchmarks demonstrate that PRISM achieves state-of-the-art fidelity in noisy depth synthesis. Critically, in downstream robotic manipulation tasks, our method facilitates a 92.1% average success rate in the real world, marking a 53.2% absolute improvement over raw simulation.

## 2. Related Work

### 2.1. The Perceptual Gap in Sim2Real Transfer

While domain randomization has successfully bridged the physical dynamics gap (Tobin et al., 2017), the perceptual observation gap remains a formidable bottleneck. Unlike RGB textural shifts (Sadeghi & Levine, 2017), the depth domain suffers from *structural geometric distortions*—ranging from stochastic fluctuations to complete signal loss—that fundamentally corrupt the geometric reasoning required for manipulation (Zhang et al., 2024). To mitigate this, prevailing solutions adopt a policy-centric online paradigm, employing adaptive environments (Yu et al., 2025) or Real2Sim restoration (Liu et al., 2025). However, these methods typically depend on auxiliary adaptation modules during deployment, introducing inference latency and suffering from generalization bottlenecks due to tight policy coupling. Conversely, our data-centric offline sensor modeling embeds real-world noise characteristics directly into training data. This enables standard policies to learn intrinsic robustness against sensor artifacts, facilitating seamless zero-shot transfer without incurring any online computational overhead.

### 2.2. Paradigms of Depth Noise Modeling

Prior efforts can be categorized into two paradigms, each with distinct limitations that motivate our PRISM.

**1) Explicit Analytical Modeling** reconstructs noise via rigorous physics. Early works like DepthSynth (Planche et al., 2017) and recent differentiable simulations (Planche & Singh, 2021; Zhang et al., 2023b) use ray-tracing to reproduce optical phenomena like multi-path interference. While rigorous, they are computationally prohibitive and rely on inaccessible proprietary sensor specifications. Lightweight alternatives like GazeboDR (Borrego et al., 2018) employ procedural injection, adding stochastic perturbations. However, these handcrafted heuristics lack physical fidelity, failing to capture material-dependent patterns (Cai et al., 2024).

**2) Implicit Data-Driven Modeling** leverages generative models to bypass analytical complexity. Techniques range from GAN-based translation (Hoffman et al., 2018) and geometry-aware synthesis (Shen et al., 2022) to recent diffusion refinement (Xu et al., 2025). Notably, DiffuDep-Grasp (Zhou et al., 2025) utilizes diffusion models for grasping-oriented noise synthesis. While visually realistic, these methods typically treat noise as a **monolithic black-box process**. By indiscriminately conflating distinct error sources—such as specular signal loss versus transparent measurement bias—into a unified stochastic variable, they risk hallucinating geometric artifacts that contradict the physical scene. Although physics-informed denoising exists (Zhang et al., 2023c), it targets restoration rather than synthesis. In contrast, PRISM introduces a physics-grounded disentanglement. We explicitly decompose noise into *Sensing Invalidation* and *Measurement Inaccuracy* based on physically motivated origins, utilizing semantic priors to guide precise synthesis that transcends the black-box limitations of monolithic baselines.

### 2.3. Visual Foundation Models as Semantic Priors

Current 3D VFMs have transcended simple depth estimation (Wang et al., 2025a), evolving into generalist geometric learners via massive-scale pre-training. State-of-the-art architectures like Metric3Dv2 (Hu et al., 2024) and MoGe (Wang et al., 2025b) employ ViT-based encoders to distill invariant geometric representations. Historically, fusing semantics with geometry has proven effective for perception, as demonstrated by early works like SIGNet (Meng et al., 2019) which utilized semantic instances to aid geometry learning. Modern VFMs implicitly encode richer semantic-physical correspondences, learning to associate visual contexts with intrinsic physical behaviors (Lin et al., 2025). In this work, instead of treating latent features merely as geometric descriptors for reconstruction, we leverage VFMs as physics reasoners for sensor modeling of artifacts based on scene semantics. To the best of our knowledge, PRISM is the first to leverage VFM-derived priors to guide physics-grounded noise disentanglement, enabling the synthesis of material-aware artifacts that elude purely monolithic translation or stochastic generation baselines.

## 3. Methodology

We present PRISM, a tripartite framework that synthesizes realistic depth by disentangling sensor noise into physically grounded modalities. As illustrated in Fig.2, PRISM follows a causal pipeline: *Reason → Disentangle → Generate*. We first formalize the problem and then detail each component.

***Problem Formulation*** Let $\mathcal{D}_s = \{(I_i, D_i^{sim})\}_{i=1}^N$ denote a synthetic dataset containing RGB images and ideal simulation depths, and $\mathcal{D}_r$ denote a real-world dataset. The goal of sensor simulation is to learn a mapping $\mathcal{M} : D^{sim} \to D^{real}$ such that policies trained on $\mathcal{M}(D^{sim})$ generalize to $\mathcal{D}_r$. Unlike prior works that model noise as a monolithic distribution $P(D^{real}|D^{sim})$, we formulate real-world depth corruption as a composition of two distinct physical modalities: **Sensing Invalidation** and **Measurement Inaccuracy**. Mathematically, we define the corrupted depth $D^{syn}$ as:

$$D^{syn}(\mathbf{x}) = \underbrace{(1 - M(\mathbf{x}))}_{\text{Validity Mask}} \cdot \left( \underbrace{D^{sim}(\mathbf{x}) + R(\mathbf{x})}_{\text{Corrupted Signal}} \right), \quad (1)$$

where $M(\mathbf{x}) \in \{0, 1\}$ is the binary invalidation mask (1 indicates invalidation), and $R(\mathbf{x}) \in \mathbb{R}$ is the measurement residual. Our objective is to learn the conditional distributions $P(M|I, D^{sim})$ and $P(R|D^{sim}, M)$ via physics-grounded disentanglement.

### 3.1. Semantic-Physics Reasoner

Formally, given an input RGB image $I \in \mathbb{R}^{3 \times H \times W}$, SPR maps its semantics to a set of multi-scale channel attention vectors $\mathcal{S} = \{\mathbf{S}_l\}_{l=1}^L$, where $L = 4$. The architecture consists of three sequential modules. First, a **Frozen 3D-VFM Backbone** extracts patch tokens $\Phi_{\text{VFM}}(I)$ that encode implicit semantic-physical properties. Second, a **Global Context Encoder** aggregates these tokens into a compact semantic embedding $\mathbf{z}_{sem}$ via Global Average Pooling (GAP) followed by a non-linear projection:

$$\mathbf{z}_{sem} = W_2 \cdot \sigma\left(W_1 \cdot \text{GAP}(\Phi_{\text{VFM}}(I))\right), \quad (2)$$

where $\sigma$ denotes SiLU activation, and $W_1, W_2$ are learnable weights designed to distill salient semantic contexts. Finally, a **Multi-Scale Semantic Attention** module projects this global context into scale-specific residual prompts $\mathbf{a}_l$ to guide the downstream disentanglement:

$$\mathbf{S}_l = \tanh\left(W_{att}^{(l)} \cdot \mathbf{z}_{sem}\right) \in [-1, 1]^{C_l}, \quad (3)$$

where $W_{att}^{(l)} \in \mathbb{R}^{C_l \times D'}$ maps the semantic embedding to the channel dimension $C_l$ of the $l$-th decoder stage.

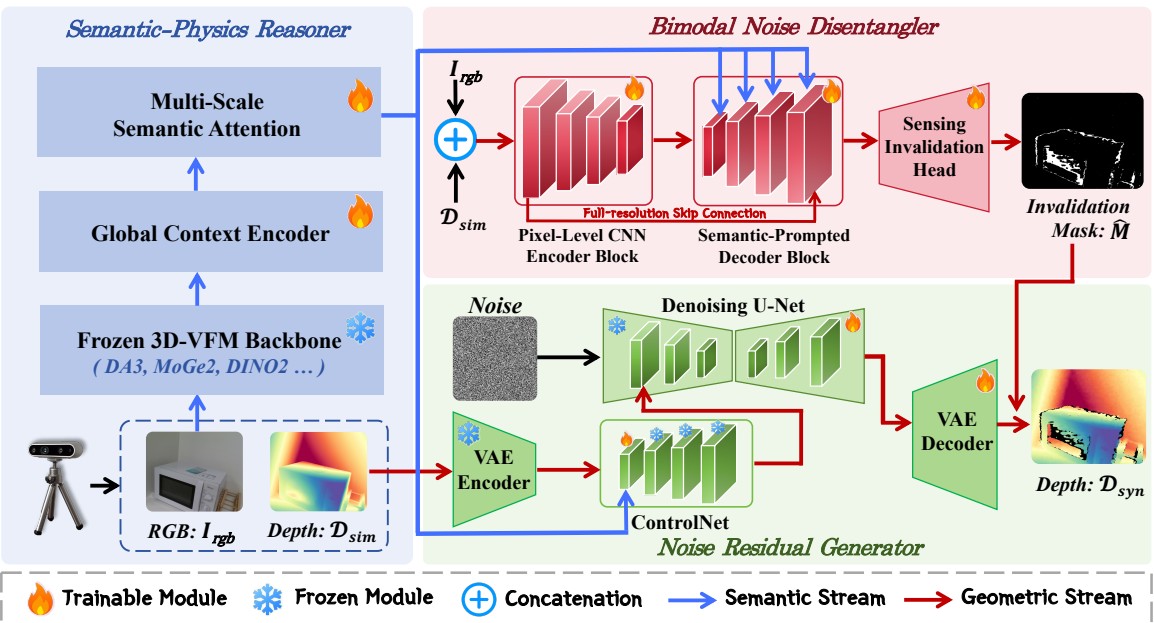

*Figure 2.* **The overview of PRISM framework.** The PRISM orchestrates three synergistic modules: the **SPR** extracts material-aware priors, the **BND** predicts sensing invalidation, and the **NRG** synthesizes measurement residuals. The diagram highlights the cross-stream modulation, where the *Semantic Stream* (from VFM) implicitly guides the *Geometric Stream* to enable physics-grounded noise synthesis.

### 3.2. Bimodal Noise Disentangler

The BND maps the concatenated RGB-Depth input $\mathbf{X} = [I; D^{sim}] \in \mathbb{R}^{4 \times H \times W}$ to a pixel-wise sensing invalidation probability map $\hat{M} \in [0, 1]^{H \times W}$. The architecture is structured into three processing stages:

*1) Fine-grained CNN Encoder.* We construct a feature pyramid $\mathcal{F} = \{\mathbf{F}_l\}_{l=0}^{4}$, where $l$ indicates the downsampling ratio $2^l$. Notably, we retain the full-resolution feature $\mathbf{F}_0$ before the first pooling stem. This preserves primitive geometric cues (e.g., thin edges) typically lost in deep downsampling, enabling precise boundary detection.

*2) Semantic-Prompted Decoder.* The decoder reconstructs resolution from coarse ($l = 4$) to fine ($l = 0$). To fuse physical semantics, we inject SPR priors $\mathbf{S}_l$ into the geometric stream via channel-wise residual scaling. Formally, at stage $l$, the feature $\mathbf{D}_l$ is computed as:

$$\mathbf{D}_l = \text{Conv}(\text{Concat}(\mathbf{D}_{l+1}^{\uparrow}, \mathbf{F}_l)) \odot (1 + \mathbf{S}_l), \quad (4)$$

where $\odot$ denotes the Hadamard product, and $\mathbf{D}_{l+1}^{\uparrow}$ represents the feature upsampled from the coarser level $l + 1$. This modulation adaptively suppresses activations in physically unstable regions based on global semantic context.

*3) Sensing Invalidation Head.* Finally, the full-resolution modulated feature $\mathbf{D}_0$ is projected to the probability space via a lightweight convolution block $\Phi_{\text{head}}$:

$$\hat{M} = \text{Sigmoid}(\Phi_{\text{head}}(\mathbf{D}_0)) \in [0, 1]^{H \times W}. \quad (5)$$

This map identifies sensing invalidation regions and as a spatial attention mask ensuring that residual generation is strictly confined to valid sensor regions.

### 3.3. Noise Residual Generator

The NRG leverages a frozen VAE and Denoising U-Net with a trainable ControlNet adapter. Given the high dynamic range of depth values, we first apply a log-scale normalization $\psi(\cdot)$ to map the simulation depth $D^{sim}$ to $[-1, 1]$. A frozen VAE Encoder $\mathcal{E}$ then compresses the ground-truth residual into the latent space $\mathbf{z}_0 = \mathcal{E}(\psi(R_{gt})) \in \mathbb{R}^{4 \times h \times w}$.

*1) Semantic-Conditioned ControlNet.* To govern the generation process, we employ a ControlNet that accepts the normalized simulation depth as the spatial condition. Crucially, to enforce physical consistency, we inject the global semantic embedding $\mathbf{z}_{sem}$ from SPR into the ControlNet. This is achieved by projecting $\mathbf{z}_{sem}$ into the time-embedding vector $\mathbf{t}_{emb}$, serving as a global style prompt:

$$\mathbf{c}_{ctl} = \text{ControlNet}(\psi(D^{sim}), \mathbf{t}_{emb} + W_{cond} \cdot \mathbf{z}_{sem}). \quad (6)$$

*2) Mask-Aware Denoising Objective.* The frozen U-Net $\epsilon_\theta$ predicts the noise $\epsilon$ added to the latent $\mathbf{z}_t$, guided by the multi-scale control signals $\mathbf{c}_{ctl}$. To prevent the model from hallucinating artifacts in invalid regions (holes), we utilize the invalidation mask $\hat{M}$ predicted by the BND module as a *hard validity constraint*. The training objective is masked to compute gradients strictly within valid sensor regions:

$$\mathcal{L}_{\text{NRG}} = \mathbb{E}_{\mathbf{z}_0, t, \epsilon} \left[ \|\epsilon - \epsilon_\theta(\mathbf{z}_t, t, \mathbf{c}_{ctl})\|_2^2 \odot (1 - \hat{M}) \right], \quad (7)$$

where $\odot$ denotes element-wise multiplication. During inference, the predicted latent is decoded by the VAE Decoder $\mathcal{D}$ to obtain the residual $\hat{R} = \psi^{-1}(\mathcal{D}(\hat{\mathbf{z}}_0))$, which is added to $D^{sim}$ to form the final synthetic depth $D^{syn}$.

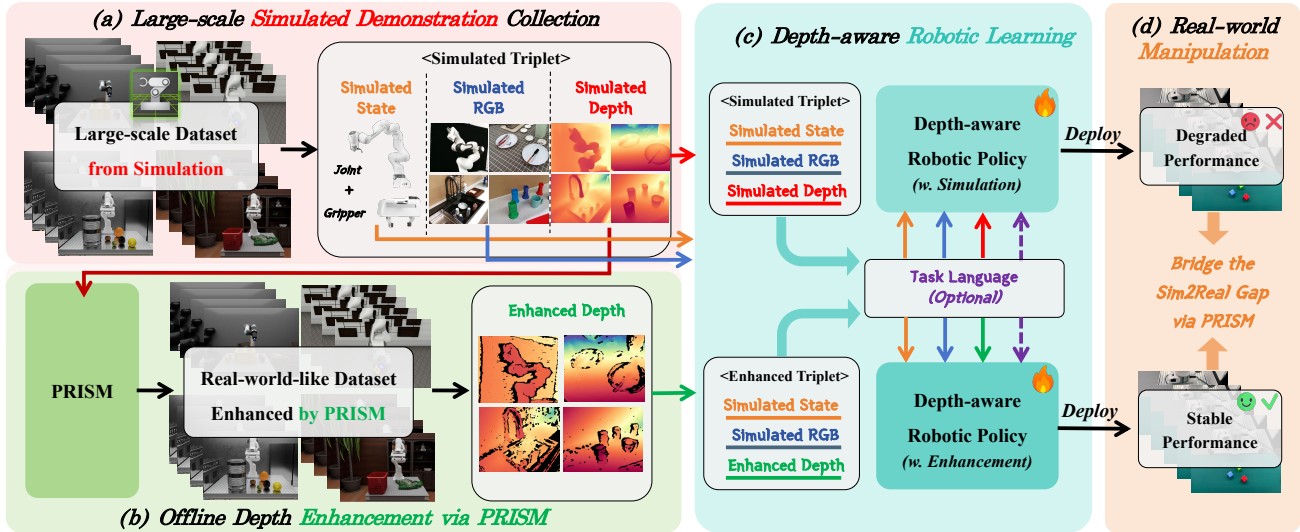

*Figure 3.* **Overview of the PRISM-enabled Sim-to-Real Pipeline.** (a) Simulation Data Collection: Large-scale expert demonstrations are collected in a simulator. (b) Offline Depth Enhancement: PRISM acts as a plug-in enhancer, transforming ideal simulation depth into real-world-like observations with physically grounded artifacts. (c) Depth-Aware Policy Learning: A modified robotic policy, equipped with a dedicated depth encoder, is trained on the enhanced dataset to learn robustness against sensor corruption. (d) Zero-Shot Real-World Transfer: The trained policy is directly deployed on physical robots, demonstrating stable manipulation without online domain adaptation.

### 3.4. Hierarchical Positive-Prioritized Supervision

To address the extreme class imbalance and ensure precise boundary detection, we propose a supervision strategy comprised of three coupled mechanisms.

*1) Multi-Scale Positive-Weighted BCE.* We apply multi-scale supervision across all decoder scales $l \in \{0, \ldots, 4\}$. To counteract the sparsity of invalid regions, we employ a positive-weighted BCE that assigns higher weights to positive samples $\mathcal{P}_l$ to penalize false negatives. For a pixel $\mathbf{x}$ at scale $l$, the loss is:

$$
\ell_{bce}^{(l)}(\mathbf{x}) = - \Big[ w_{pos} \cdot M_{gt}(\mathbf{x}) \log \hat{M}_l(\mathbf{x}) \\
+ (1 - M_{gt}(\mathbf{x})) \log(1 - \hat{M}_l(\mathbf{x})) \Big],
\tag{8}
$$

where we set $w_{pos} = 3.0$ to significantly boost recall for rare sensor invalidations, as configured in our training.

*2) Positive-Preserving Dynamic Hard Mining.* To prevent dominant background gradients from overwhelming fine-grained artifacts, we propose PP-DHM. It dynamically constructs the active backpropagation set $\Omega_l$ by retaining *all* positive samples $\mathcal{P}_l$ and mining only the hardest fraction of negatives $\mathcal{N}_{hard}$ (see Appendix for pseudocode):

$$
\Omega_l = \mathcal{P}_l \cup \text{TopK}(\mathcal{N}_l, \gamma),
\tag{9}
$$

where the mining ratio $\gamma(t)$ decays linearly from $0.25$ to $0.1$ over training epochs $t$, progressively focusing on the most challenging pixels around boundaries.

*3) Sequential Optimization Objectives.* Since PRISM is trained in two stages, we define separate objectives for the disentangler and generator.

*Stage I: Noise Disentanglement Learning.* In addition to the pixel-wise classification, we introduce a Dice Loss to enforce shape compactness and prevent trivial solutions. The BND is optimized to minimize the weighted sum of these objectives over the mined sets:

$$
\mathcal{L}_{\text{BND}} = \sum_{l=0}^{4} \lambda_l \left( \sum_{\mathbf{x} \in \Omega_l} \ell_{bce}^{(l)}(\mathbf{x}) + \lambda_{dice} \mathcal{L}_{dice}(\hat{M}_l, M_{gt}) \right),
\tag{10}
$$

where $\lambda_0 = 1.0$ for the main output, $\lambda_{1,2,3,4} = 0.5$ for auxiliary scales and $\lambda_{dice} = 0.3$ for shape consistency.

*Stage II: Residual Generation Learning.* The NRG is trained with the frozen BND. Its objective is the mask-constrained loss restricted to valid regions, shown in Eq.(7).

## 4. Application

To validate the efficacy of PRISM in downstream robotic tasks, we establish a full-cycle Sim-to-Real pipeline covering simulation construction, data enhancement, and real-world deployment (see Fig. 3).

***Data Generation Pipeline.*** We construct object-centric simulation environments aligned with real-world setups via differentiable-rendering-based camera calibration. To efficiently acquire large-scale expert behaviors, we leverage the MimicGen (Mandlekar et al., 2023) integrated with whole-body control(Haviland et al., 2022), collecting a dataset of ideal RGB-D demonstrations $\mathcal{D}_{sim}$. Crucially, PRISM functions as an **offline sensor simulator**: it processes the clean depth maps in $\mathcal{D}_{sim}$ to generate a physically rigor-

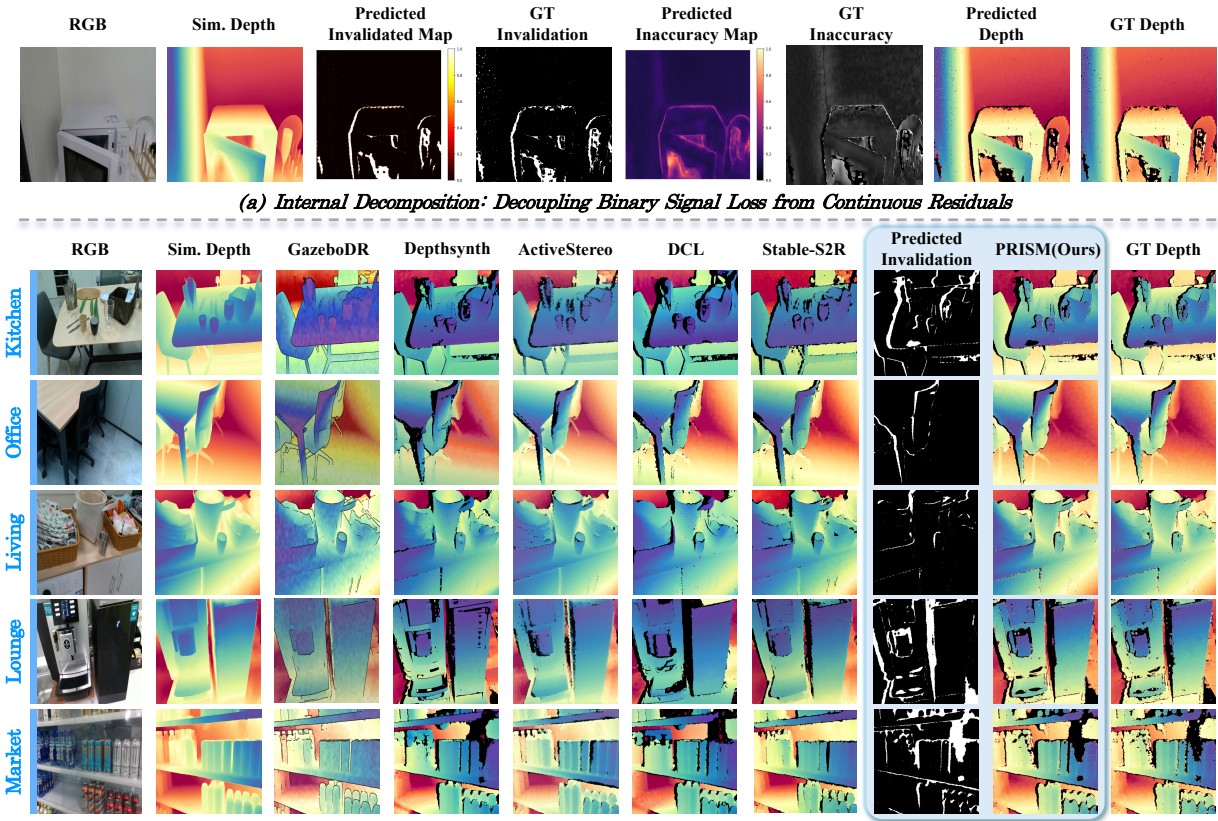

(a) *Internal Decomposition: Decoupling Binary Signal Loss from Continuous Residuals*

(b) *Visual Benchmarking: Qualitative Comparison against SOTA Baselines in Complex Scenes*

*Figure 4.* **Qualitative evaluation of physics-grounded depth synthesis.** (a) Disentangled Generation: PRISM decomposes noise into *Sensing Invalidation* (binary masks) and *Measurement Inaccuracy* (continuous residuals) to compose final depth. (b) Baseline Comparison: Across diverse scenes involving specular and transparent objects, PRISM precisely localizes artifacts to physically unstable regions.

ous, corrupted dataset $\mathcal{D}_{real-like}$ before training begins. By implicitly modeling sensing invalidation and measurement inaccuracy, PRISM embeds real-world sensor characteristics into the training depth without incurring computational overhead during policy inference.

***Policy Learning & Zero-Shot Transfer.*** We formulate the imitation learning objective as learning a policy $\pi_\theta(\mathcal{A}^t|\mathcal{O}^t)$ that maps observations to action sequences. Unlike point-cloud approaches (Ze et al., 2024; Wang et al., 2024) that require aggressive artifact removal, we advocate for a direct **Depth-Aware Policy**. We modify the standard visual backbone by introducing a dual-stream ResNet encoder(Lu et al., 2020)—one for RGB and a dedicated branch for Depth—fused via late concatenation. The policy is trained entirely on the PRISM-enhanced dataset $\mathcal{D}_{real-like}$. During real-world deployment, this enables **zero-shot transfer**: having "seen" structural corruptions during training, the policy exhibits intrinsic robustness to physical sensor noise, generalizing seamlessly to real robots without online adaptation. Comprehensive details on network architecture, and hyper-parameters are provided in Appendix.

## 5. Experiment

### 5.1. Experimental Settings

***Implementation Details.*** Implemented in PyTorch on 8 NVIDIA H200 GPUs, PRISM trains on $512^2$ images with a batch size of 32. The BND leverages a frozen MoGe2-Large backbone, optimized via AdamW. The NRG adapts an SD 2.1 backbone initialized with a frozen OpenCLIP encoder. Training proceeds in two phases totaling 100 epochs: first freezing the VAE and U-Net to optimize only the Control-Net, followed by joint fine-tuning where the U-Net and VAE decoder are unfrozen (VAE encoder frozen). Inference utilizes DDIM sampling (50 steps, 9.0 scale).

***Datasets.*** We train on ByteCameraDepth (Liu et al., 2025), comprising real-world raw depth captured from 7 cameras across 10 modes in 7 diverse scene categories. We utilize the provided models to construct aligned sim-real pairs and define the **ground-truth invalidation mask** by identifying native sensor failures (zeros or NaNs) in raw depth maps. Furthermore, we conduct zero-shot evaluation on NYU-Depth-v2 (Silberman et al., 2012) to assess the generalization in cross-domain of Kinect camera.

*Table 1.* **Quantitative comparison of depth synthesis fidelity on ByteCameraDepth (In-Domain of Realsense D435).** We evaluate three aspects: Overall Metrics (global reconstruction quality), Sensing Invalidation (detection of sensor failures), and Measurement Inaccuracy (precision in valid regions). *Note*: Baselines marked with '-' do not support separate invalidation masks from continuous noise.

| Method | Overall Metrics | | | | Sensing Invalidation | | | Meas. Inaccuracy | | |
|---|---|---|---|---|---|---|---|---|---|---|
| | MAE $\downarrow$ | RMSE $\downarrow$ | AbsRel $\downarrow$ | $\delta < 1.25 \uparrow$ | IoU $\uparrow$ | F1 $\uparrow$ | Recall $\uparrow$ | MAE $\downarrow$ | RMSE $\downarrow$ | Acc-$\delta \uparrow$ |
| GazeboDR | 0.203 | 0.258 | 0.167 | 0.724 | - | - | - | 0.178 | 0.224 | 68.3 |
| DepthSynth | 0.115 | 0.143 | 0.092 | 0.884 | 0.651 | 0.789 | 0.734 | 0.094 | 0.121 | 86.1 |
| ActiveStereo | 0.094 | 0.118 | 0.075 | 0.916 | 0.806 | 0.893 | 0.858 | 0.078 | 0.101 | 89.4 |
| DCL-Depth | 0.098 | 0.125 | 0.078 | 0.908 | 0.785 | 0.879 | 0.845 | 0.082 | 0.106 | 88.6 |
| Stable-Sim2Real | 0.087 | 0.106 | 0.069 | 0.931 | 0.876 | 0.934 | 0.912 | 0.070 | 0.091 | 91.7 |
| **PRISM (Ours)** | **0.076** | **0.085** | **0.054** | **0.964** | **0.952** | **0.975** | **0.991** | **0.052** | **0.068** | **95.3** |

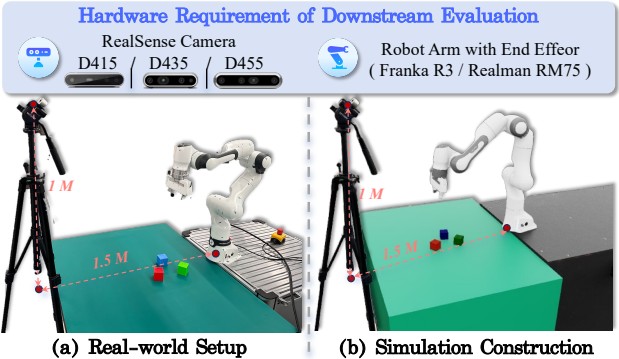

*Figure 5.* **Hardware Requirement of Downstream Evaluation.** The real-world setup consists of two platforms: Franka Research 3 for specular and deformable manipulation tasks, and Realman RM75 for long-horizon articulation tasks.

***Evaluation Metrics.*** To comprehensively assess depth synthesis quality and downstream robustness (see Appendix B.4 for formal definitions), we evaluate three complementary aspects. *Overall Fidelity*: MAE, RMSE, and threshold accuracy $\delta < 1.25$ measure global reconstruction quality. *Sensing Invalidation*: IoU, F1-Score, and Recall over predicted hole masks, where high Recall is safety-critical as missed voids directly cause policy failures. *Measurement Inaccuracy*: valid-region MAE and Acc-$\delta$ measure surface precision. For downstream manipulation, we report *Task Success Rate (SR)* per 20 trials, averaged across 3 random seeds.

***Baselines.*** We benchmark against five methods across two paradigms, all adapted for 1-channel 16-bit depth.

*1) Explicit Analytical Modeling:* We select **GazeboDR** (Borrego et al., 2018) as the representative method of predefined heuristics, injecting procedural perturbations into depth. We also include **DepthSynth** (Planche et al., 2017) as a classic general-purpose synthesis framework, and **ActiveStereo** (Zhang et al., 2023b), representing the SOTA in physics-grounded optical simulation.

*2) Implicit Data-Driven Modeling:* We evaluate Differential Contrastive Learning (**DCL-Depth**) (Shen et al., 2022) as the SOTA representative for geometry-aware image translation. Finally, we compare against **Stable-Sim2Real** (Xu et al., 2025), the current SOTA generative framework for depth refinement with two cascade diffusion models.

*Table 2.* **Cross-camera synthesis generalization.** Models trained on ByteCameraDepth are directly tested on NYU-Depth-v2 (**Zero-shot in Kinect Camera**) to assess robustness.

| Method | Overall Fidelity | | Sensing Inv. | |
|---|---|---|---|---|
| | MAE $\downarrow$ | $\delta < 1.25 \uparrow$ | IoU $\uparrow$ | F1 $\uparrow$ |
| GazeboDR | 0.248 | 0.678 | - | - |
| DepthSynth | 0.138 | 0.832 | 0.542 | 0.703 |
| ActiveStereo | 0.112 | 0.878 | 0.685 | 0.813 |
| DCL-Depth | 0.118 | 0.865 | 0.658 | 0.794 |
| Stable-Sim2Real | 0.102 | 0.894 | 0.756 | 0.861 |
| **PRISM (Ours)** | **0.082** | **0.941** | **0.923** | **0.960** |

## 5.2. Depth Fidelity Evaluation

**Physics-Consistent Synthesis & Fidelity.** Qualitative results (Fig.4) show that while baselines like DCL-Depth and Stable-Sim2Real often suffer from mode collapse or global hallucination, PRISM precisely localizes physically meaningful artifacts. Guided by SPR, our method correctly targets specular and transparent surfaces for invalidation, consequently avoiding indiscriminate background corruption to preserve valid geometric structures. This physics alignment translates to SOTA quantitative performance on ByteCameraDepth (Tab.1). Specifically, our *Hierarchical Positive-Prioritized Supervision* yields high Invalidation IoU/F1 by overcoming mask sparsity, while the disentangled NRG achieves the lowest valid-region MAE/RMSE, effectively preventing the averaging effect inherent in monolithic noise modeling.

**Zero-Shot Generalization.** Zero-shot evaluation on the NYU-Depth-v2 dataset (Tab.2) demonstrates PRISM's superior robustness under noisy domain shifts when compared to overfitting-prone baselines. By leveraging generic VFM priors rather than fitting specific dataset statistics, PRISM effectively reasons about noise generation through material-correlated visual patterns rather than superficial domain textures. Consequently, this mechanism enables consistent synthesis across previously unseen scenarios with minimal performance degradation, firmly validating its applicability as a general-purpose sensor simulator.

*Table 3.* **Multi-task manipulation success rates with RISE policy.** We evaluate 6 tasks with diverse properties: Specular (☀), Deformable (🖊), Articulated (⚙), and Long-Horizon (⏱). Results are reported as mean ± std across 3 random seeds (/20 trials).

| Task | Properties | | | | Explicit Analytical Modeling | | | Implicit Data-Driven Modeling | | |
| --- | --- | --- | --- | --- | --- | --- | --- | --- | --- | --- |
| | ☀ | 🖊 | ⚙ | ⏱ | GazeboDR | DepthSynth | ActiveStereo | DCL-Depth | Stable-S2R | **PRISM** |
| Can-to-Plate | ✔ | | | | $18.7_{\pm0.6}$ | $\mathbf{20.0_{\pm0.0}}$ | $\mathbf{20.0_{\pm0.0}}$ | $\mathbf{20.0_{\pm0.0}}$ | $\mathbf{20.0_{\pm0.0}}$ | $\mathbf{20.0_{\pm0.0}}$ |
| Banana-to-Plate | | ✔ | | | $18.0_{\pm1.0}$ | $19.3_{\pm0.6}$ | $19.7_{\pm0.6}$ | $19.7_{\pm0.6}$ | $\mathbf{20.0_{\pm0.0}}$ | $\mathbf{20.0_{\pm0.0}}$ |
| Stack Cubes | ✔ | | | ✔ | $12.7_{\pm0.6}$ | $14.3_{\pm0.6}$ | $16.7_{\pm0.6}$ | $16.3_{\pm1.5}$ | $18.7_{\pm0.6}$ | $\mathbf{19.3_{\pm0.6}}$ |
| Bucket-to-Plate | ✔ | | ✔ | | $11.7_{\pm0.6}$ | $13.7_{\pm1.2}$ | $15.7_{\pm0.6}$ | $15.3_{\pm1.2}$ | $17.7_{\pm0.6}$ | $\mathbf{17.7_{\pm0.6}}$ |
| Toyball-to-Drawer | ✔ | ✔ | ✔ | ✔ | $4.7_{\pm0.6}$ | $7.7_{\pm0.6}$ | $11.7_{\pm0.6}$ | $11.3_{\pm1.2}$ | $16.3_{\pm0.6}$ | $\mathbf{17.3_{\pm1.2}}$ |
| Bread-to-Microwave | ✔ | ✔ | ✔ | ✔ | $3.7_{\pm0.6}$ | $6.3_{\pm0.6}$ | $10.3_{\pm0.6}$ | $10.7_{\pm0.6}$ | $16.3_{\pm0.6}$ | $\mathbf{16.7_{\pm0.6}}$ |
| **Average Success** | - | - | - | - | 57.8% | 67.8% | 78.3% | 77.8% | 90.8% | **92.5%** |

*Table 4.* **Cross-policy Sim2Real generalization.** We report the average success rate (%) across all 6 tasks using three different policy architectures. PRISM consistently outperforms baselines.

| Baselines | DP3 | PPT | RISE | Average |
| --- | --- | --- | --- | --- |
| Sim + Clean | 35.3 | 38.3 | 43.1 | 38.9 |
| Sim + GazeboDR | 49.2 | 53.3 | 57.8 | 53.4 |
| Sim + DepthSynth | 60.6 | 66.4 | 67.8 | 64.9 |
| Sim + ActiveStereo | 75.6 | 80.6 | 78.3 | 78.2 |
| Sim + DCL-Depth | 74.2 | 78.6 | 77.8 | 76.9 |
| Sim + Stable-Sim2Real | 86.7 | 89.7 | 90.8 | 89.1 |
| **Sim + PRISM (Ours)** | **91.1** | **92.8** | **92.5** | **92.1** |

## 5.3. Downstream Application Evaluation

***Tasks & Policies Adaptation.*** We establish a benchmark of 6 diverse manipulation tasks (Tab.3) across two robotic platforms to evaluate challenging physical properties. Precision tasks on a Franka arm require policies to overcome fragmented boundaries and surface distortions for specular cans, deformable bananas, and reflective cubes (T1-T3). Articulated tasks on a Realman RM75 arm test robustness against translucent refraction, transparent obstacles, and severe occlusions during long-horizon behaviors like interacting with a drawer or microwave (T4-T6). Detailed configurations are in Appendix C.4. To assess the universality of our method, we adapt three state-of-the-art 3D policy architectures—3D Diffusion Policy (**DP3**) (Ze et al., 2024), **PPT** (Hua et al., 2024), and **RISE** (Wang et al., 2024)—into depth-aware policies that directly process 16-bit depth.

***Multi-Task Sim2Real Manipulation.*** Table 3 reports real-world success rates using the RISE backbone. PRISM achieves a commanding lead, particularly in tasks involving *Toyball-to-Drawer* and *Bread-to-Microwave*, where analytical simulations fail to reproduce the surface property-correlated invalidation gap. By physically grounding noise synthesis, PRISM forces the policy to learn compliant behaviors robust to sensor dropouts (e.g., inferring geometry from boundaries), achieving an average success rate of **92.5%** and significantly outperforming all analytical and generative simulation baselines.

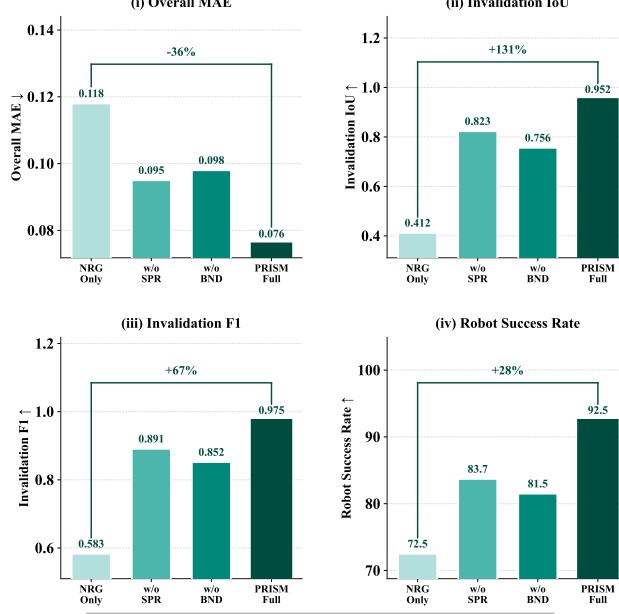

*Figure 6.* **Components Contribution.** Assessing the impact of removing SPR and BND on Invalidation IoU and Robot SR.

***Cross-Policy Generalization.*** As shown in Tab.4, PRISM consistently boosts Sim2Real performance across all three policy architectures. Notably, the performance gains are not isolated to a single policy type; the absolute improvement remains robust at about 3% over the strongest generative baseline regardless of whether the policy relies on 3D DP, PPT, or RISE. This uniform enhancement confirms that PRISM functions as a fundamental data-level sensor simulator. By effectively closing the geometric sensing gap independently of the specific downstream control policy, it provides a universally applicable solution for deployment.

## 5.4. Ablation Studies

We validate the efficacy of PRISM's causal design—*Reason → Disentangle → Generate*—through comprehensive ablations on the ByteCameraDepth dataset. We summarize the contributions of system components, VFM backbones, and supervision strategies in the following sections.

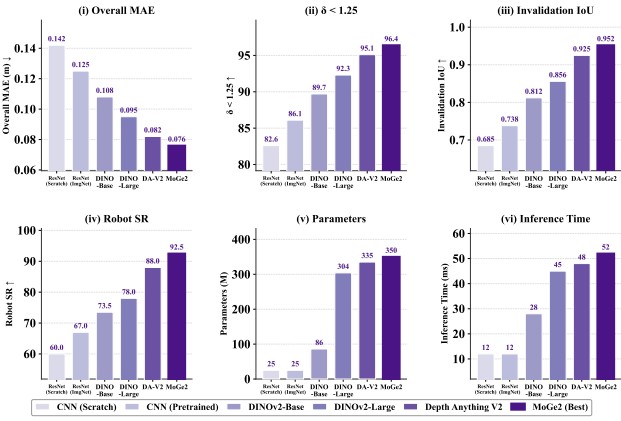

*Figure 7.* **Semantics Efficacy.** Comparing generic vs. geometric priors. 3D VFMs show superior material awareness.

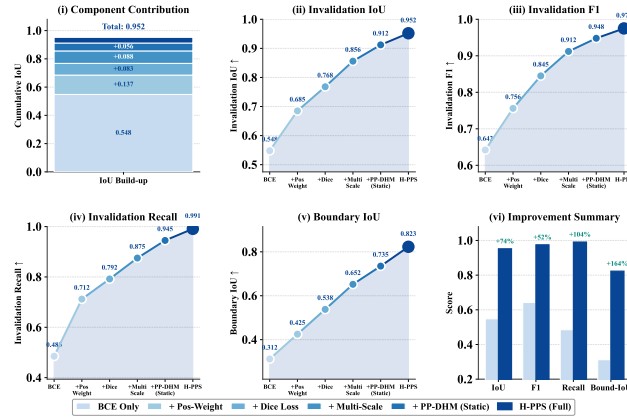

*Figure 8.* **Supervision Strategy.** Progressive improvement by adding Positive-Weighting, Multi-Scale, and PP-DHM.

***Impact of Causal Architecture.*** We treat the diffusion-based NRG as the backbone and selectively remove components (Fig. 6). 1) NRG Only (Baseline): The pure baseline, equivalent to removing both SPR and BND, exhibits the lowest robotic success rate, proving that physical grounding is essential for Sim2Real transfer. 2) w/o SPR (Physics-Blind): Removing the VFM-driven priors leads to a sharp performance drop specifically on non-Lambertian surfaces. This confirms that without semantic reasoning, the model struggles to infer physics-dependent sensor invalidation. 3) w/o BND (Monolithic): Bypassing the disentangler forces the NRG to learn a conflated noise distribution. This results in the most significant degradation in overall MAE and severe hallucinations in holes, validating that invalidation modeling is crucial to prevent the averaging effect.

***Efficacy of Physics Priors.*** PRISM leverages a frozen VFM to extract semantic-physical priors. We compare generic vision backbones against geometry-aware models (Fig. 7). Results indicate that 3D VFMs (e.g., MoGe2, Depth Anything V2) significantly outperform generic ViTs (DINOv2) and CNNs (ResNet). Notably, MoGe2 achieves the highest Robot SR with larger parameter overhead. This suggests pre-trained geometric foundation models implicitly encode robust semantics-physics correlations, which are more transferable to sensor simulation than purely semantic features.

***Analysis of Supervision Strategy.*** We dissect the H-PPS via progressive ablation (Fig. 8). 1) Class Imbalance: Standard BCE fails to capture sparse invalidation masks (especially when $< 7.5\%$ pixels). Adding multi-scale positive-weighted drastically boosts *recall*, preventing the model from collapsing to a all-valid trivial solution. 2) Shape & Boundary: Integrating Dice Loss improves the compactness of predicted holes, while PP-DHM is critical for boundary precision. Compared to static mining, our dynamic decay strategy $\gamma(t)$ stabilizes training and improves Boundary-IoU by 8.8% (12.0% relative), ensuring sharp delineation of sensor artifacts.

## 6. Conclusion

We present PRISM, a framework that unveils the physical causality of depth sensor noise via disentanglement. By orchestrating VFM-driven reasoning with physics-grounded generation, our work demonstrates that embedding physics-semantics into noise modeling, rather than relying on black-box randomization, is the key to bridge the geometric perceptual gap, paving the way for trustworthy embodied AI.

## 7. Limitations

While PRISM demonstrates strong capabilities in simulated-depth enhancement, it possesses certain limitations. First, its sensor-specific training constrains cross-sensor generalizability: the modules are optimized for a fixed sensor family and degrade under out-of-distribution sensing principles, motivating lightweight sensor adaptation. Second, the current per-frame generation pipeline does not explicitly enforce temporal consistency for highly dynamic scenes, leaving flickering noise across frames as a key open challenge. Finally, our formulation focuses on intrinsic surface-correlated noise patterns, leaving interaction-induced extrinsic noise and domain-shifted extreme optical environments for future exploration. Details are provided in Appendix F.

## Acknowledge

This work was supported in part by the New Generation Artificial Intelligence-National Science and Technology Major Project (Grant No. 2025ZD0122901), alongside the Fundamental and Interdisciplinary Disciplines Breakthrough Plan of the Ministry of Education of China, the funding from the Shanghai Committee of Science and Technology, China (Grant No. 24511103200), the Shanghai Artificial Intelligence Laboratory, and the XPLORER PRIZE. The authors also sincerely appreciate the crucial resources and support provided by the Ant Group Research Fund.

## Impact Statement

This paper presents work whose goal is to advance the field of Machine Learning. There are many potential societal consequences of our work, none which we feel must be specifically highlighted here.

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

# Appendix Overview

This appendix provides comprehensive details to support the reproducibility and theoretical grounding of PRISM. The content is organized as follows:

- **Sec. A**: Detailed architecture specifications and algorithmic pseudocode for PRISM and H-PPS.

- **Sec. B**: Comprehensive experimental settings, baseline configurations, and dataset descriptions.

- **Sec. C**: The full-cycle Sim-to-Real pipeline, including environment construction and policy adaptation.

- **Sec. D**: Extended quantitative ablations, sensitivity analyses, and cross-domain evaluations.

- **Sec. E**: Additional qualitative visualizations of depth synthesis and robotic manipulation.

- **Sec. F**: Discussion on limitations and broader societal impact.

## A. Implementation Details and Algorithms

We provide comprehensive implementation details to ensure full reproducibility of PRISM. This section covers background on diffusion-based generation (§A.1), detailed architecture specifications (§A.2), algorithmic pseudocode (§A.3), and training hyperparameters (§A.4).

### A.1. Background: Diffusion Models and ControlNet

**Denoising Diffusion Probabilistic Models (DDPM).** Our NRG builds upon the DDPM framework (Ho et al., 2020), which models data distribution $p(\mathbf{x})$ via a Markov chain. The *forward process* gradually corrupts data $\mathbf{x}_0$ by adding Gaussian noise:

$$q(\mathbf{x}_t|\mathbf{x}_0) = \mathcal{N}(\mathbf{x}_t; \sqrt{\bar{\alpha}_t}\mathbf{x}_0, (1 - \bar{\alpha}_t)\mathbf{I}), \quad t \in \{1, \ldots, T\} \tag{11}$$

where $\bar{\alpha}_t = \prod_{s=1}^t \alpha_s$ and $\{\alpha_t\}$ defines the noise schedule. The *reverse process* learns to denoise via a neural network $\epsilon_\theta$:

$$p_\theta(\mathbf{x}_{t-1}|\mathbf{x}_t) = \mathcal{N}\left(\mathbf{x}_{t-1}; \frac{1}{\sqrt{\alpha_t}}\left(\mathbf{x}_t - \frac{1 - \alpha_t}{\sqrt{1 - \bar{\alpha}_t}}\epsilon_\theta(\mathbf{x}_t, t)\right), \sigma_t^2\mathbf{I}\right) \tag{12}$$

Training minimizes the simplified objective: $\mathcal{L}_{\text{simple}} = \mathbb{E}_{t,\mathbf{x}_0,\epsilon}\left[\|\epsilon - \epsilon_\theta(\mathbf{x}_t, t)\|^2\right]$.

**ControlNet for Conditional Generation.** ControlNet (Zhang et al., 2023a) extends pre-trained diffusion models by creating a trainable copy of the encoder blocks that accept additional spatial conditions $\mathbf{c}$. The control signals are injected into the frozen U-Net decoder via zero-convolution layers:

$$\mathbf{y}_c = \mathcal{F}(\mathbf{x}; \Theta) + \mathcal{Z}(\mathcal{F}(\mathbf{x} + \mathcal{Z}(\mathbf{c}; \Theta_{z1}); \Theta_c); \Theta_{z2}) \tag{13}$$

where $\mathcal{Z}(\cdot; \cdot)$ denotes zero-convolution (initialized to zero weights), $\Theta$ represents frozen parameters, and $\Theta_c$ represents trainable ControlNet parameters. This design enables precise spatial control while preserving the generative capacity of the base model.

### A.2. Architecture Specifications

PRISM comprises three synergistic modules following a causal design: *Reason → Disentangle → Generate*. We detail the layer-wise specifications for each component below.

#### A.2.1. SEMANTIC-PHYSICS REASONER (SPR)

The SPR is responsible for extracting material-aware priors. To preserve global semantic consistency without introducing high-frequency noise from patch tokens, we employ a Global Average Pooling (GAP) strategy. Table 5 details the specific dimensions of the MLP projection heads and the multi-scale attention mechanisms that adapt the global semantic embedding $\mathbf{z}_{sem}$ to different resolutions.

*Table 5.* SPR Architecture Specification. The VFM backbone is frozen during training.

| Module | Input Dim | Output Dim | Description |
|---|---|---|---|
| VFM Backbone (MoGe2-L) | $[B, 3, H, W]$ | $[B, N, 1024]$ | MoGe2-Large, frozen $N = \frac{H}{14} \times \frac{W}{14}$ |
| Global Avg Pool | $[B, N, 1024]$ | $[B, 1024]$ | Spatial aggregation |
| Context MLP | $[B, 1024]$ | $[B, 256]$ | Linear(1024→512) + SiLU + Linear(512→256) |
| Scale Attn $\ell = 1$ | $[B, 256]$ | $[B, 64]$ | Linear + Tanh |
| Scale Attn $\ell = 2$ | $[B, 256]$ | $[B, 128]$ | Linear + Tanh |
| Scale Attn $\ell = 3$ | $[B, 256]$ | $[B, 256]$ | Linear + Tanh |
| Scale Attn $\ell = 4$ | $[B, 256]$ | $[B, 512]$ | Linear + Tanh |

The multi-scale channel attention vectors $\{\mathbf{S}_\ell\}_{\ell=1}^4$ are computed as:

$$\mathbf{z}_{\text{global}} = \text{GAP}(\Phi_{\text{VFM}}(\mathbf{I})), \quad \mathbf{z}_{\text{sem}} = \text{MLP}(\mathbf{z}_{\text{global}}), \quad \mathbf{S}_\ell = \tanh(\mathbf{W}_\ell \cdot \mathbf{z}_{\text{sem}}) \tag{14}$$

where $\mathbf{W}_\ell \in \mathbb{R}^{C_\ell \times 256}$ projects the semantic embedding to scale-specific channel dimensions.

### A.2.2. BIMODAL NOISE DISENTANGLER (BND)

The BND predicts the pixel-wise probability of sensing invalidation. To achieve precise boundary delineation, the encoder retains full-resolution features in the initial stages. Table 6 outlines the detailed structure of the encoder, highlighting the preservation of spatial dimensions before the first downsampling stem.

*Table 6.* BND Encoder Architecture. ConvBlock consists of Conv3×3 + BatchNorm + ReLU.

| Stage | Input | Output | Resolution | Operations |
|---|---|---|---|---|
| Pre-Stem | $[B, 4, H, W]$ | $[B, 32, H, W]$ | $H \times W$ | Conv3×3 + BN + ReLU |
| Stem | $[B, 32, H, W]$ | $[B, 64, \frac{H}{2}, \frac{W}{2}]$ | $\frac{H}{2}$ | Conv3×3(s=2) + BN + ReLU |
| Stage 1 | $[B, 64, \frac{H}{2}, \frac{W}{2}]$ | $[B, 64, \frac{H}{2}, \frac{W}{2}]$ | $\frac{H}{2}$ | ConvBlock ×2 |
| Stage 2 | $[B, 64, \frac{H}{2}, \frac{W}{2}]$ | $[B, 128, \frac{H}{4}, \frac{W}{4}]$ | $\frac{H}{4}$ | ConvBlock(s=2) + ConvBlock |
| Stage 3 | $[B, 128, \frac{H}{4}, \frac{W}{4}]$ | $[B, 256, \frac{H}{8}, \frac{W}{8}]$ | $\frac{H}{8}$ | ConvBlock(s=2) + ConvBlock |
| Stage 4 | $[B, 256, \frac{H}{8}, \frac{W}{8}]$ | $[B, 512, \frac{H}{16}, \frac{W}{16}]$ | $\frac{H}{16}$ | ConvBlock(s=2) + ConvBlock |

For the decoder, we employ a semantic-prompted modulation mechanism. Instead of simple concatenation, the SPR priors modulate the skip-connected features channel-wise. Table 7 specifies the input and output dimensions for each decoder stage, along with the specific modulation operation applied.

*Table 7.* BND Decoder Architecture with Semantic-Prompted Modulation.

| Stage | Input | Skip | Output | Resolution | Modulation |
|---|---|---|---|---|---|
| Dec4 | $[B, 512, \frac{H}{16}]$ | $[B, 256, \frac{H}{8}]$ | $[B, 512, \frac{H}{8}]$ | $\frac{H}{8}$ | $\mathbf{F} \odot (1 + \mathbf{S}_4)$ |
| Dec3 | $[B, 512, \frac{H}{8}]$ | $[B, 128, \frac{H}{4}]$ | $[B, 256, \frac{H}{4}]$ | $\frac{H}{4}$ | $\mathbf{F} \odot (1 + \mathbf{S}_3)$ |
| Dec2 | $[B, 256, \frac{H}{4}]$ | $[B, 64, \frac{H}{2}]$ | $[B, 128, \frac{H}{2}]$ | $\frac{H}{2}$ | $\mathbf{F} \odot (1 + \mathbf{S}_2)$ |
| Dec1 | $[B, 128, \frac{H}{2}]$ | $[B, 32, H]$ | $[B, 128, H]$ | $H$ | Skip only |
| Head | $[B, 128, H, W]$ | — | $[B, 1, H, W]$ | $H$ | Conv + Sigmoid |

The semantic-prompted residual modulation at decoder stage $\ell$ is formulated as:

$$\mathbf{D}_\ell = \text{Conv}\left(\text{Concat}\left(\text{Upsample}(\mathbf{D}_{\ell+1}), \mathbf{F}_\ell\right)\right) \odot (1 + \mathbf{S}_\ell) \tag{15}$$

where $\mathbf{F}_\ell$ denotes skip-connected encoder features and $\mathbf{S}_\ell$ is the scale-specific channel attention from SPR.

### A.2.3. NOISE RESIDUAL GENERATOR (NRG)

NRG is built upon a conditional diffusion framework. It integrates a trainable ControlNet adapter with a frozen Stable Diffusion backbone. Table 8 clarifies which components are frozen and which are fine-tuned, ensuring the preservation of the generative prior while adapting to depth noise synthesis.

*Table 8.* NRG Architecture Specification. "Trainable" indicates whether parameters are updated during training.

| Module | Input | Output | Trainable | Description |
|---|---|---|---|---|
| VFM (MoGe2-L) | $[B, 3, H, W]$ | $[B, N, 1024]$ | ✗ | Semantic extraction |
| VFM Projector | $[B, 1024]$ | $[B, 320]$ | ✓ | MLP for conditioning |
| VAE Encoder | $[B, 3, H, W]$ | $[B, 4, \frac{H}{8}, \frac{W}{8}]$ | ✗ | Latent compression |
| VAE Decoder | $[B, 4, \frac{H}{8}, \frac{W}{8}]$ | $[B, 3, H, W]$ | ✓ | Fine-tuned decoding |
| ControlNet | $[B, 3, H, W]$ | Multi-scale | ✓ | Spatial conditioning |
| U-Net | $[B, 4, \frac{H}{8}, \frac{W}{8}]$ | $[B, 4, \frac{H}{8}, \frac{W}{8}]$ | ✓ (Stage 2) | Denoising backbone |

The configuration for the diffusion process is critical for stable training. Table 9 provides the exact parameters used for the DDPM schedule and the DDIM inference sampling, including the log-scale depth normalization range.

*Table 9.* Diffusion Configuration for NRG.

| Parameter | Value | Parameter | Value |
|---|---|---|---|
| Base Model | Stable Diffusion 2.1 | DDIM $\eta$ | 0.0 (deterministic) |
| DDPM Timesteps | 1000 (training) | Guidance Scale | 9.0 |
| DDIM Steps | 50 (inference) | Residual Scale | 2.0 |
| Depth Range | $[0.05\text{m}, 5.0\text{m}]$ | Latent Channels | 4 |

**Log-Scale Depth Normalization.** To handle the wide dynamic range of depth values, we apply log-scale normalization:

$$\psi(\mathbf{D}) = 2 \cdot \frac{\log(\mathbf{D} + \epsilon) - \log(d_{\min})}{\log(d_{\max}) - \log(d_{\min})} - 1, \quad \psi^{-1}(\mathbf{z}) = \exp\left(\frac{(\mathbf{z} + 1)}{2}(\log(d_{\max}) - \log(d_{\min})) + \log(d_{\min})\right) \quad (16)$$

where $d_{\min} = 0.05\text{m}$, $d_{\max} = 5.0\text{m}$, and $\epsilon = 10^{-6}$ for numerical stability.

### A.3. Algorithmic Description

We provide detailed pseudocode for the two core algorithms: the complete PRISM inference pipeline (Algorithm 1) and H-PPS (Algorithm 2) for training BND. Note that we use explicit syntax for loops and returns to ensure clarity.

**Key Insight of PP-DHM.** Standard Online Hard Example Mining (OHEM) discards easy samples including rare positives, causing mode collapse to all-negative predictions. In contrast, PP-DHM *always preserves all positive pixels* regardless of their loss magnitude, while only negative samples undergo hard mining. The dynamic decay $\gamma(t)$ enables broader coverage during early training and boundary-focused learning in later stages.

*Table 10.* Trainable Parameter Summary for PRISM.

| Module | Parameters | Components |
|---|---|---|
| SPR | ~0.8M | Context MLP + Scale Attention heads |
| BND | ~21M | CNN Encoder + Decoder + Prediction Heads |
| NRG | ~361M | ControlNet + VFM Projector + VAE Decoder (fine-tuned) |
| **Total** | **~382M** | |

---

**Algorithm 1** PRISM Inference Pipeline

---

**Require:** Simulated depth $\mathbf{D}^{\text{sim}}$, RGB image $\mathbf{I}$
**Require:** Frozen VFM $\Phi_{\text{VFM}}$, Trained BND $\theta_{\text{BND}}$, Trained NRG $\theta_{\text{NRG}}$
**Ensure:** Synthesized realistic depth $\mathbf{D}^{\text{syn}}$
 1: // Stage 1: Semantic-Physics Reasoning (SPR)
 2: $\mathbf{F}_{\text{VFM}} \leftarrow \Phi_{\text{VFM}}(\mathbf{I})$ {Extract VFM patch tokens, $[B, N, 1024]$}
 3: $\mathbf{z}_{\text{global}} \leftarrow \text{GAP}(\mathbf{F}_{\text{VFM}})$ {Global context, $[B, 1024]$}
 4: $\mathbf{z}_{\text{sem}} \leftarrow \text{MLP}(\mathbf{z}_{\text{global}})$ {Semantic embedding, $[B, 256]$}
 5: **for** $\ell = 1$ **to** $4$ **do**
 6: $\quad \mathbf{S}_\ell \leftarrow \tanh(\mathbf{W}_\ell \cdot \mathbf{z}_{\text{sem}})$ {Multi-scale channel attention}
 7: **end for**
 8: // Stage 2: Bimodal Noise Disentanglement (BND)
 9: $\mathbf{X} \leftarrow \text{Concat}([\mathbf{I}; \mathbf{D}^{\text{sim}}])$ {$[B, 4, H, W]$}
10: $\{\mathbf{F}_0, \ldots, \mathbf{F}_4\} \leftarrow \text{Encoder}(\mathbf{X})$ {Multi-scale features}
11: **for** $\ell = 4$ **to** $0$ **do**
12: $\quad \mathbf{D}_\ell \leftarrow \text{Conv}(\text{Concat}(\text{Upsample}(\mathbf{D}_{\ell+1}), \mathbf{F}_\ell)) \odot (1 + \mathbf{S}_\ell)$
13: **end for**
14: $\hat{\mathbf{M}} \leftarrow \sigma(\text{Head}(\mathbf{D}_0))$ {Invalidation probability, $[B, 1, H, W]$}
15: $\mathbf{M} \leftarrow \mathbb{1}[\hat{\mathbf{M}} > \tau]$ {Binary mask, $\tau = 0.5$}
16: // Stage 3: Noise Residual Generation (NRG)
17: $\mathbf{D}^{\text{sim}}_{\text{norm}} \leftarrow \psi(\mathbf{D}^{\text{sim}})$ {Log-scale normalization to $[-1, 1]$}
18: $\mathbf{c}_{\text{ctl}} \leftarrow \text{ControlNet}(\mathbf{D}^{\text{sim}}_{\text{norm}}, t_{\text{emb}} + \mathbf{W}_{\text{cond}} \cdot \mathbf{z}_{\text{sem}})$
19: $\mathbf{z}_T \sim \mathcal{N}(\mathbf{0}, \mathbf{I})$ {Initial noise}
20: **for** $t = T$ **to** $1$ **do**
21: $\quad \hat{\boldsymbol{\epsilon}} \leftarrow \text{UNet}(\mathbf{z}_t, t, \mathbf{c}_{\text{ctl}})$
22: $\quad \mathbf{z}_{t-1} \leftarrow \text{DDIM\_Step}(\mathbf{z}_t, \hat{\boldsymbol{\epsilon}}, t)$ {50-step DDIM}
23: **end for**
24: $\hat{\mathbf{R}} \leftarrow \psi^{-1}(\text{VAE\_Decoder}(\mathbf{z}_0))$ {Decode residual}
25: // Stage 4: Depth Composition
26: $\mathbf{D}^{\text{syn}} \leftarrow (1 - \mathbf{M}) \odot (\mathbf{D}^{\text{sim}} + \hat{\mathbf{R}})$ {Apply disentangled noise}
27: **return** $\mathbf{D}^{\text{syn}}$

---

### A.4. Training Hyperparameters

We train PRISM using $8\times$ NVIDIA H200 GPUs with PyTorch 2.1 and mixed-precision (FP16) training. The BND module is trained for 100 epochs with learning rate $1 \times 10^{-4}$ and AdamW optimizer. The NRG module follows a two-stage training strategy: Stage 2 trains ControlNet for 50 epochs (lr=$1 \times 10^{-4}$), followed by Stage 3 joint fine-tuning for 50 epochs (lr=$1 \times 10^{-5}$). Finally, Table 10 aggregates the parameter counts for all modules, illustrating the model's efficiency relative to its generative capabilities.

## B. Experimental Configuration

This section provides comprehensive experimental settings to ensure reproducibility. We detail the implementation configurations (§B.1), dataset preparation (§B.2), baseline implementations (§B.3), and evaluation metrics definitions (§B.4).

### B.1. Implementation Details

PRISM is implemented using the PyTorch framework and optimized for high-performance computing clusters. To ensure stable convergence and efficient training, we carefully tuned the hyperparameters based on pilot studies on the validation set. Table 11 summarizes the comprehensive hardware specifications and training parameters. Notably, we employ a batch size of 32 per GPU across 8 H200 GPUs to stabilize the gradient estimation for the diffusion model.

The optimization objective involves multiple loss components tailored for sparsity and geometric consistency. Table 12 details the specific weights assigned to each loss term. We assign a high weight ($w_{\text{pos}} = 3.0$) to the positive class in the

---

**Algorithm 2** Hierarchical Positive-Prioritized Supervision (H-PPS)

---

**Require:** Predicted masks $\{\hat{\mathbf{M}}_\ell\}_{\ell=0}^{L}$ at $L+1$ scales, GT mask $\mathbf{M}_{\text{gt}}$
**Require:** Current epoch $t$, total epochs $T$, hyperparameters: $w_{\text{pos}}, \lambda_{\text{dice}}, \gamma_{\text{start}}, \gamma_{\text{end}}$
**Ensure:** Total BND loss $\mathcal{L}_{\text{BND}}$

1: $\mathcal{L}_{\text{BND}} \leftarrow 0$
2: **for** $\ell = 0$ **to** $L$ **do**
3:    $\mathbf{M}_{\text{gt}}^{(\ell)} \leftarrow \text{Downsample}(\mathbf{M}_{\text{gt}}, \text{scale} = 2^\ell)$ {Multi-scale GT}
4:    // **Step 1:** Compute per-pixel Positive-Weighted BCE
5:    **for** each pixel $\mathbf{x}$ **do**
6:       $\ell_{\text{bce}}(\mathbf{x}) \leftarrow - \Big[ w_{\text{pos}} \cdot \mathbf{M}_{\text{gt}}(\mathbf{x}) \cdot \log(\hat{\mathbf{M}}_\ell(\mathbf{x})) + (1 - \mathbf{M}_{\text{gt}}(\mathbf{x})) \cdot \log(1 - \hat{\mathbf{M}}_\ell(\mathbf{x})) \Big]$
7:    **end for**
8:    // **Step 2:** Positive-Preserving Dynamic Hard Mining (PP-DHM)
9:    $\mathcal{P} \leftarrow \{\mathbf{x} : \mathbf{M}_{\text{gt}}(\mathbf{x}) = 1\}$ {All positive pixels}
10:   $\mathcal{N} \leftarrow \{\mathbf{x} : \mathbf{M}_{\text{gt}}(\mathbf{x}) = 0\}$ {All negative pixels}
11:   $\gamma(t) \leftarrow \gamma_{\text{start}} - (\gamma_{\text{start}} - \gamma_{\text{end}}) \times (t/T)$ {Dynamic decay}
12:   $\mathcal{N}_{\text{hard}} \leftarrow \text{TopK}(\mathcal{N}, k = \lceil |\mathcal{N}| \times \gamma(t) \rceil, \text{key} = \ell_{\text{bce}})$ {Hardest negatives}
13:   $\Omega_\ell \leftarrow \mathcal{P} \cup \mathcal{N}_{\text{hard}}$ {**Preserve ALL positives**}
14:   // **Step 3:** Compute scale-specific loss
15:   $\mathcal{L}_{\text{bce}}^{(\ell)} \leftarrow \frac{1}{|\Omega_\ell|} \sum_{\mathbf{x} \in \Omega_\ell} \ell_{\text{bce}}(\mathbf{x})$
16:   $\mathcal{L}_{\text{dice}}^{(\ell)} \leftarrow 1 - \frac{2 \sum \hat{\mathbf{M}}_\ell \cdot \mathbf{M}_{\text{gt}}^{(\ell)} + \epsilon}{\sum \hat{\mathbf{M}}_\ell + \sum \mathbf{M}_{\text{gt}}^{(\ell)} + \epsilon}$ {Dice loss}
17:   $\lambda_\ell \leftarrow 1.0$ **if** $\ell = 0$ **else** $0.5$ {Main vs auxiliary}
18:   $\mathcal{L}_{\text{BND}} \leftarrow \mathcal{L}_{\text{BND}} + \lambda_\ell \cdot (\mathcal{L}_{\text{bce}}^{(\ell)} + \lambda_{\text{dice}} \cdot \mathcal{L}_{\text{dice}}^{(\ell)})$
19: **end for**
20: **return** $\mathcal{L}_{\text{BND}}$

---

H-PPS loss to explicitly penalize missed detections of sparse sensor invalidations, ensuring high recall.

**Progressive Training Strategy.** To align the conditional generation with physical constraints, we adopt a two-stage training strategy for the NRG module:

- **Stage 1 (Adapter Alignment, Epochs 1-50):** We freeze the U-Net backbone and VAE Encoder, training only the ControlNet adapter and VFM Projector with a learning rate of $1 \times 10^{-4}$. This phase allows the model to learn the spatial mapping between simulation depth and residual features without disrupting the pre-trained generative prior.

- **Stage 2 (End-to-End Refinement, Epochs 51-100):** We unlock the VAE Decoder and fine-tune it jointly with the ControlNet using a reduced learning rate of $1 \times 10^{-5}$. This refinement sharpens the high-frequency details of the synthesized noise artifacts.

### B.2. Dataset Preparation

#### B.2.1. BYTECAMERADEPTH DATASET

To ensure our model generalizes across diverse sensing principles, we utilize the ByteCameraDepth dataset. Unlike previous datasets confined to a single sensor type, ByteCameraDepth captures real-world depth data using a heterogeneous array of consumer-grade sensors, covering the three dominant depth sensing technologies: Active Stereo, Time-of-Flight (ToF), and LiDAR. Table 13 provides a breakdown of the dataset splits used in our experiments.

**Sensor Heterogeneity & Scene Diversity.** The dataset features high diversity in both sensor characteristics and environmental contexts.

- **Sensor Modalities:** The inclusion of Active Stereo (prone to multipath interference), ToF (sensitive to reflectivity), and LiDAR (sparse but accurate) ensures that PRISM learns a sensor-agnostic representation of "physical invalidation" rather than overfitting to specific hardware artifacts.

*Table 11.* PRISM Training Configuration. All experiments use mixed-precision (FP16) training with DDP.

| Category | Parameter | Value |
|---|---|---|
| **Hardware** | GPU | $8\times$ NVIDIA H200 |
| | Framework | PyTorch 2.1 + CUDA 12.1 |
| | Distributed | DDP (DistributedDataParallel) |
| | Precision | FP16 (AMP enabled) |
| **Common** | Image Resolution | $512 \times 512$ |
| | Optimizer | AdamW ($\beta_1$=0.9, $\beta_2$=0.999) |
| | Random Seed | 42 |
| **BND Module** | Batch Size (per GPU) | 32 |
| | Learning Rate | $1 \times 10^{-4}$ |
| | Weight Decay | 0.01 |
| | LR Schedule | Cosine Annealing |
| | Training Epochs | 100 |
| | Gradient Clip | 1.0 |
| **NRG Module** | Batch Size (per GPU) | 32 |
| | Learning Rate | $1 \times 10^{-4}$ (Stage 2), $1 \times 10^{-5}$ (Stage 3) |
| | Weight Decay | $1 \times 10^{-5}$ |
| | Training Epochs | 100 (50 Stage 2 + 50 Stage 3) |
| | Warmup Epochs | 3 per stage |

*Table 12.* Loss Function Hyperparameters for H-PPS and NRG.

| Module | Loss Component | Weight | Description |
|---|---|---|---|
| **BND (H-PPS)** | Positive-Weighted BCE | $\lambda = 1.0$ | Main supervision |
| | Positive Weight | $w_{\text{pos}} = 3.0$ | Recall enhancement |
| | Dice Loss | $\lambda_{\text{dice}} = 0.3$ | Shape regularization |
| | Multi-Scale Auxiliary | $\lambda_{\text{aux}} = 0.5$ | Deep supervision (4 scales) |
| | OHEM Start Ratio | $\gamma_{\text{start}} = 0.25$ | 25% hardest negatives |
| | OHEM End Ratio | $\gamma_{\text{end}} = 0.10$ | 10% at final epoch |
| **NRG** | L1 Reconstruction | $\lambda = 1.0$ | Valid regions only |
| | Hole Mask Weight | 0.0 | No gradient in holes |

- **Material Properties:** The scenes are curated to include challenging objects that typically cause sensor failure, including transparent objects (glass bottles, beakers), specular surfaces (metal cans, polished steel), absorptive materials (black rubber, fabric), and fine geometries (wires, mesh).

**Data Preprocessing & GT Extraction.** The raw data is processed to generate aligned Simulation-Real pairs.

1. **Ground-Truth Invalidation Mask:** We define the "ground truth" for invalidation not by manual annotation, but by leveraging the sensor's native failure flags. A pixel is marked as invalid ($\mathbf{M}_{\text{gt}} = 1$) if the raw sensor output is zero, NaN, or exceeds the maximum reliable range ($d > 5.0$m). This provides a physically accurate supervision signal for signal loss.

2. **Normalization:** Raw depth values are converted to a log-scale representation to balance the gradients between near-field and far-field regions.

3. **Alignment:** Sim-Real pairs are spatially registered using high-precision factory calibration parameters.

*Table 13.* ByteCameraDepth Dataset Statistics.

| Split | Samples | Scene Instances | Usage |
|---|---|---|---|
| Training | 45,000 | 39 | Model training |
| Validation | 5,000 | 39 | Hyperparameter tuning |
| Test | 5,000 | 39 | Final evaluation |
| **Total** | **55,000** | **39** | |

### B.2.2. NYU-DEPTH-V2 ZERO-SHOT EVALUATION

To rigorously evaluate cross-domain generalization, we perform zero-shot testing on the NYU-Depth-v2 dataset (Silberman et al., 2012). This dataset presents a significant domain shift as it was captured with an older generation Structured-Light sensor not seen during training.

*Table 14.* NYU-Depth-v2 Dataset Configuration for Zero-Shot Evaluation.

| Property | Value |
|---|---|
| Sensor Type | Legacy Structured-Light Sensor |
| Test Split | 654 images (Standard Eigen Split) |
| Depth Range | 0.1m – 10.0m |
| Resolution | $640 \times 480$ (Center Cropped) |

Table 14 details the configuration for this evaluation. We use the standard 654-image test split and evaluate on the center crop to assess how well PRISM transfers learned physical priors to unseen sensor architectures and environments.

### B.3. Baseline Implementation

To demonstrate the efficacy of PRISM, we benchmark against five representative methods categorized into two paradigms: *Explicit Analytical Modeling* and *Implicit Data-Driven Modeling*. All baselines are adapted to predict 1-channel 16-bit depth maps and are trained/tuned on the ByteCameraDepth dataset for fair comparison.

### B.3.1. EXPLICIT ANALYTICAL MODELING

Methods in this category rely on predefined physical heuristics or ray-tracing simulations. They do not learn from data distributions but instead attempt to analytically reconstruct the noise formation process.

**1) GazeboDR (Borrego et al., 2018) (Predefined Heuristics).**
*Mechanism:* A standard domain randomization technique used in robotic simulators (e.g., Gazebo). It injects procedural noise into clean simulation depth.
*Adaptation:* We implemented a composite noise model comprising Gaussian noise ($\sigma \sim \mathcal{U}[0.01, 0.05]$), random signal dropout ($p \sim \mathcal{U}[0.05, 0.15]$), and edge erosion kernels. The parameters were tuned to statistically match the noise magnitude of the training set.
*Pros/Cons:* Computationally free but lacks physical realism; noise is spatially independent of scene semantics.

**2) DepthSynth (Planche et al., 2017) (Ray-Tracing Simulation).**
*Mechanism:* A classic general-purpose synthesis framework that simulates the sensor's optical path. It uses ray-tracing to model the structured-light projection and stereo matching pipeline.
*Adaptation:* We configured the simulator with a standard baseline (9-pixel) and used the scene's mesh material properties to approximate reflectance.
*Pros/Cons:* Captures geometric occlusion well but struggles with complex material interactions (e.g., sub-surface scattering) due to simplified BRDF models.

**3) ActiveStereo (Zhang et al., 2023b) (SOTA Physics-Grounded).**
*Mechanism:* The current state-of-the-art in physics-based simulation. It explicitly simulates the IR emitter pattern, light transport, and the sensor's disparity estimation logic, including multi-path interference modeling.

*Adaptation:* We used the official implementation, calibrating the virtual sensor parameters (focal length, baseline) to match the real-world setup.
*Pros/Cons:* Highly accurate for specific sensor models but computationally prohibitive and generalizes poorly to sensors with unknown internal logic (e.g., proprietary ToF filters).

### B.3.2. IMPLICIT DATA-DRIVEN MODELING

Methods in this category leverage deep neural networks to learn the mapping from clean simulation to noisy real depth directly from data.

**4) DCL-Depth (Shen et al., 2022) (Geometry-Aware Translation).**
*Mechanism:* Originally designed for unpaired depth completion, we adapted it as a Sim-to-Real translator. It uses a U-Net architecture with a contrastive loss to maintain geometric structure while transferring domain style.
*Adaptation:* We trained the model on paired ByteCameraDepth data for 100 epochs. The loss function combines L1 reconstruction with a patch-based contrastive loss ($\tau = 0.07$).
*Pros/Cons:* Preserves global geometry better than GANs but models noise as a monolithic texture, often failing to predict sharp invalidation boundaries.

**5) Stable-Sim2Real (Xu et al., 2025) (SOTA Generative Framework).**
*Mechanism:* The current SOTA generative approach employing two cascaded diffusion models: one for coarse residual prediction and another for fine-grained refinement using ControlNet.
*Adaptation:* We trained the framework on our dataset, using Stable Diffusion 2.1 as the backbone. We followed the official two-stage training protocol (Residual Stage $\rightarrow$ Refinement Stage).
*Pros/Cons:* Generates highly realistic textures but suffers from the "black-box" issue—it hallucinates artifacts in safe regions and lacks explicit control over signal loss (invalidation), leading to lower robotic safety.

### B.4. Evaluation Metrics Definition

To provide a holistic assessment of synthesized depth quality, we employ a diverse set of metrics covering three distinct aspects: *Overall Quality*, *Sensing Invalidation* (Safety), and *Measurement Inaccuracy* (Precision).

### B.4.1. OVERALL DEPTH RECONSTRUCTION

These metrics assess the global fidelity of the synthesized depth $\hat{\mathbf{D}}$ against the ground truth real depth $\mathbf{D}$. They reflect the general visual similarity across the entire image.

- **Mean Absolute Error (MAE):** Measures the average magnitude of absolute errors.

$$\text{MAE} = \frac{1}{N} \sum_{i \in \Omega} |\mathbf{D}_i - \hat{\mathbf{D}}_i| \tag{17}$$

- **Root Mean Square Error (RMSE):** Penalizes large errors more heavily than MAE, making it highly sensitive to outliers or extreme noise spikes.

$$\text{RMSE} = \sqrt{\frac{1}{N} \sum_{i \in \Omega} (\mathbf{D}_i - \hat{\mathbf{D}}_i)^2} \tag{18}$$

- **Absolute Relative Error (AbsRel):** Normalizes the absolute error by the ground-truth depth, making the metric scale-invariant and suitable for comparing errors across near-field and far-field regions.

$$\text{AbsRel} = \frac{1}{N} \sum_{i \in \Omega} \frac{|\mathbf{D}_i - \hat{\mathbf{D}}_i|}{\mathbf{D}_i} \tag{19}$$

- **Threshold Accuracy ($\delta < \tau$):** The percentage of pixels where the ratio between prediction and ground truth is within a threshold $\tau$. This is a standard robust metric for depth estimation, indicating the fraction of "reliable" predictions. We report $\delta < 1.25, 1.25^2, 1.25^3$.

$$\delta_{<\tau} = \frac{1}{N} \sum_{i \in \Omega} \not\Vdash \left[ \max\left( \frac{\mathbf{D}_i}{\hat{\mathbf{D}}_i}, \frac{\hat{\mathbf{D}}_i}{\mathbf{D}_i} \right) < \tau \right] \tag{20}$$

### B.4.2. SENSING INVALIDATION (SAFETY CRITICAL)

These metrics evaluate the model's ability to correctly predict *where* the sensor will fail (i.e., return 0 or NaN). This is critical for robotic safety, as blind spots must be identified to avoid collisions.

- **Intersection over Union (IoU):** Measures the overlap between the predicted invalidation mask $\hat{\mathbf{M}}$ and the ground truth mask $\mathbf{M}_{gt}$. A high IoU indicates precise spatial localization of sensor holes.

$$\text{IoU} = \frac{\text{TP}}{\text{TP} + \text{FP} + \text{FN}} \tag{21}$$

- **Recall (Sensitivity):** The fraction of real invalid pixels that are correctly detected.

$$\text{Recall} = \frac{\text{TP}}{\text{TP} + \text{FN}} \tag{22}$$

*Significance:* In robotics, Recall is paramount. Missing a hole (False Negative) means the robot treats empty space as valid geometry, potentially leading to grasp failures or collisions.

- **F1-Score:** The harmonic mean of Precision and Recall, providing a balanced view of performance given the extreme class imbalance (invalidation regions typically cover $< 10\%$ of pixels).

$$\text{F1} = \frac{2 \cdot \text{TP}}{2 \cdot \text{TP} + \text{FP} + \text{FN}} \tag{23}$$

### B.4.3. MEASUREMENT INACCURACY (PRECISION)

These metrics focus exclusively on the valid regions (where $\mathbf{M}_{gt} = 0$). They measure how accurately the model simulates the metric distortion (noise) on visible surfaces.

- **MAE-Valid / RMSE-Valid:** Computed strictly on pixels where the sensor returns a valid signal. Let $\Omega_{valid} = \{i \mid \mathbf{M}_{gt,i} = 0\}$.

$$\text{MAE}_{valid} = \frac{1}{|\Omega_{valid}|} \sum_{i \in \Omega_{valid}} |\mathbf{D}_i - \hat{\mathbf{D}}_i| \tag{24}$$

*Significance:* By isolating valid regions, these metrics verify that the model generates realistic measurement noise (e.g., fluctuations on surfaces) without conflating it with the massive errors caused by signal loss. This distinction proves the efficacy of our disentanglement strategy, ensuring that improvement in one modality does not come at the cost of other.

## C. Robotic Sim-to-Real Pipeline

In this section, we elaborate on the full-cycle pipeline deployed to validate PRISM's effectiveness in real-world robotic manipulation. We describe the simulation environment construction (§C.1), the large-scale demonstration generation pipeline (§C.2), the adaptation of policy architectures (§C.3), and the detailed definitions of the manipulation tasks (§C.4).

### C.1. Simulation Environment Construction

We construct high-fidelity digital twin environments using Isaac Sim 5.0.0, leveraging its PhysX 5 engine for stable contact dynamics. As illustrated in Fig. 9 (*Hardware Requirement of Downstream Evaluation*), our setup involves two distinct robotic platforms to evaluate manipulation capabilities across different scales and kinematic structures:

- **Platform A (Precise Manipulation):** A Franka Research 3 (7-DoF) arm equipped with a parallel jaw gripper. This platform is dedicated to Tasks 1–3 (fine-grained pick-and-place), prioritizing high-precision end-effector control.

- **Platform B (Complex Articulation):** A Realman RM75 (6-DoF) dual-arm setup (using a single arm for current tasks) designed for Tasks 4–6, involving larger workspaces and interaction with articulated objects (drawers, microwaves).

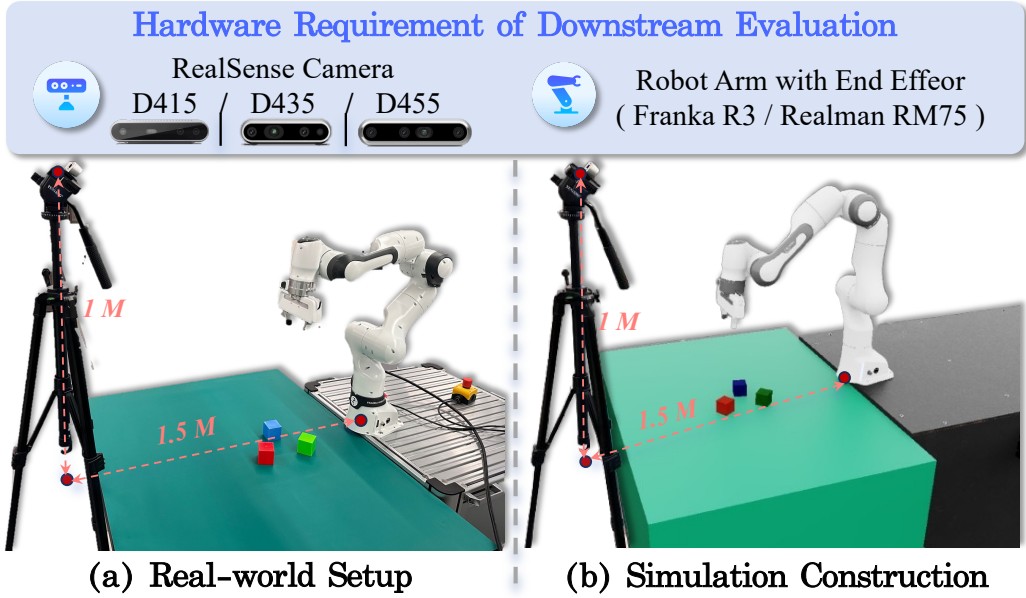

*Figure 9.* **Hardware Requirement of Downstream Evaluation.** The real-world setup consists of two platforms: Franka Research 3 for precision tasks, and Realman RM75 for long-horizon articulation tasks. We utilize RealSense D435 cameras for RGB-D perception.

**Aligned Sim-Real Pairs.** A critical prerequisite for our data-centric sensor modeling is the geometric alignment between simulation and reality. We adopt an *Object-Centric Alignment* strategy:

1. **Geometric Alignment:** We import centimeter-accurate CAD models of the robots and manipulation objects (YCB objects and articulated furniture). The camera extrinsics are calibrated using a differentiable-rendering-based optimization method to minimize the re-projection error between simulated markers and real-world keypoints.

2. **Visual Decoupling:** While geometry is strictly aligned, we do *not* enforce strict photorealism for the background. Instead, PRISM learns to handle background variations via the semantic consistency inherent in the VFM priors. This allows us to use simple PBR materials in simulation while deploying in complex real-world backgrounds.

### C.2. Large-Scale Demonstration Collection

To scale up data collection without expensive human teleoperation, we employ a WBC-MimicGen pipeline. This combines human demonstration augmentation with whole-body control for kinematic feasibility.

**1) Source Demonstration Collection.** We collect a small set ($N = 10$) of high-quality human demonstrations for each task using a VR teleoperation interface. These seed trajectories capture the key semantic milestones (e.g., grasping pose, pre-insertion alignment).

**2) MimicGen Augmentation.** We leverage MimicGen (Mandlekar et al., 2023) to procedurally generate thousands of new variations. By determining valid object initial poses and target regions, MimicGen synthesizes new task instances by transforming the source end-effector poses to align with the new object configurations.

**3) Whole-Body Control (WBC) Smoothing.** Direct interpolation of end-effector poses often leads to jerky joint motions or kinematic singularities. We integrate a Whole-Body Controller (Haviland et al., 2022) to solve for joint velocities $\dot{\mathbf{q}}$ that satisfy the task-space objectives $\dot{\mathbf{x}}_{ref}$ while respecting joint limits and collision constraints:

$$\dot{\mathbf{q}} = J^{\dagger}\dot{\mathbf{x}}_{ref} + (I - J^{\dagger}J)\dot{\mathbf{q}}_{null} \tag{25}$$

This ensures that the generated dataset $\mathcal{D}_{sim}$ contains smooth, physically feasible trajectories ideal for policy learning.

## C.3. Policy Architectures and Adaptation

To evaluate the universality of PRISM, we adapt three state-of-the-art policy architectures—**RISE**, **3D Diffusion Policy (3D DP)**, and **PPT**—to directly process 16-bit depth maps. Note that while the original implementations often rely on point clouds or RGB-only inputs, we modify their encoders to support a **Depth-Aware Dual-Stream** mechanism.

**Dual-Stream ResNet Encoder.** We replace the standard visual backbone with a dual-stream architecture:

- **RGB Stream:** A ResNet-50 initialized with ImageNet weights processes the $512 \times 512 \times 3$ RGB image.

- **Depth Stream:** A modified ResNet-18 processes the $512 \times 512 \times 1$ depth map. The first convolution layer is modified to accept 1-channel input. The depth values are normalized to $[-1, 1]$ using the same log-scale function $\psi(\cdot)$ in PRISM.

The feature maps from both streams are flattened and concatenated before being projected into the policy's latent dimension.

**Specific Adaptations:**

- **RISE (Transformer-based):** RISE (Wang et al., 2024) originally employs a sparse 3D encoder to process point clouds. We adapt it by replacing the sparse encoder with our dual-stream ResNet. The flattened RGB-D features are projected into tokens and fed into the RISE Transformer backbone, utilizing its diffusion head for action prediction. The depth stream effectively substitutes the explicit point cloud geometry with dense depth embeddings.

- **3D Diffusion Policy (DP3, Point-based):** DP3 (Ze et al., 2024) is designed to condition diffusion models on 3D point cloud representations. To adapt it for our image-space pipeline (2D-DP3), we replace the pointnet encoder with our dual-stream visual encoder. The fused RGB-D embeddings serve as the global conditioning context for the conditional U-Net, allowing the policy to retain geometric sensitivity without requiring explicit point cloud construction.

- **PPT (Transformer-based):** The Proprioceptive Pointcloud Transformer (PPT) (Hua et al., 2024) typically fuses proprioception with point cloud tokens. We modify the visual tokenizer to accept patch tokens from our dual-stream encoder instead of point clusters. The depth-stream tokens provide the critical local geometry information that PPT originally derived from 3D coordinates, ensuring the Transformer attention layers can still reason about spatial structure.

## C.4. Task Descriptions

We evaluate PRISM on 6 distinct manipulation tasks, rigorously categorized by their dominant physical properties (Specular, Transparent, Deformable, Articulated) and the robotic platform utilized.

**Platform A Tasks (Precision Manipulation on Franka Research 3).** These tasks require high-precision end-effector control to handle objects with severely corrupted geometries.

- **T1: Can-to-Plate (Specular).** *Goal:* Pick a metallic coke can and place it upright on a plate. *Success Criterion:* The can remains stable on the plate for $> 2$ seconds. *Physical Challenge:* The can's surface exhibits high reflectivity, causing the raw depth sensor to return zero values (holes) at the center of the cylinder. The policy must infer the object's centroid from the fragmented boundary information.

- **T2: Banana-to-Plate (Deformable).** *Goal:* Grasp a yellow banana and place it on a plate. *Success Criterion:* The banana rests on the plate without slipping or falling. *Physical Challenge:* The organic geometry is irregular, and subsurface scattering on the skin surface creates measurement noise (inaccuracy) rather than complete signal loss, testing the policy's robustness to metric distortions.

- **T3: Stack Cubes (Specular + Multi-Stage).** *Goal:* A sequential stacking task involving three cubes made of *specular opaque plastic*. First, stack a red cube onto a blue base cube, then stack a green cube onto the red one. *Success Criterion:* The three-cube tower remains stable without toppling for $> 5$ seconds. *Physical Challenge:* This task demands extreme vertical precision. The plastic's mirror-like surface creates "ghost" depth readings and holes on the top surfaces. Successfully stacking three cubes requires robustly overcoming these artifacts to prevent cumulative geometric errors, where even slight misalignment in the first stage leads to failure in the final stack.

**Platform B Tasks (Articulated Interaction on Realman RM75).** These tasks involve larger workspaces and interaction with mechanisms, introducing challenges related to thin structures and transparent obstacles.

- **T4: Bucket-to-Plate (Specular + Translucent + Articulated).** *Goal:* Grasp a *translucent plastic bucket* by its *articulated handle*, lift it into the air, and accurately place it onto a porcelain plate. *Success Criterion:* The bucket is stably positioned within the plate's rim. *Physical Challenge:* This task combines optical and dynamic complexity. The bucket's body is translucent, inducing complex refraction and subsurface scattering (Measurement Inaccuracy) that distort its perceived shape. Furthermore, the smooth plastic surface also produces specular reflections under direct lighting, causing partial Sensing Invalidation (holes) at highlight regions—making this task a dual-modality noise challenge. Additionally, the *articulated handle* introduces under-actuated dynamics; the policy must stabilize the swinging bucket during transport despite the noisy depth feedback from its semi-transparent body.

- **T5: Toyball-to-Drawer (Specular + Articulated + Multi-Stage).** *Goal:* Open a drawer via its handle, pick a ball, place it inside, and close the drawer. *Success Criterion:* The drawer is fully closed with the ball inside. *Physical Challenge:* A long-horizon task where the drawer handle is reflective. Crucially, during the insertion phase, the robot's arm occludes the sensor, requiring the policy to maintain a robust geometric representation under occlusion.

- **T6: Bread-to-Microwave (Transparent + Articulated).** *Goal:* Open a microwave door, place a bread slice inside, and close the door. *Success Criterion:* The door is latched closed with the bread inside. *Physical Challenge:* This is the most challenging task. The microwave door features a semi-transparent grid window that generates chaotic depth artifacts (refraction and ghosting). Standard policies often crash into the door, perceiving it as empty space (invalidation). PRISM-trained policies learn to grasp the opaque edge of the handle, treating the noisy window as a "no-go" zone.

## D. Additional Quantitative Analysis

In this section, we provide the detailed numerical data for the ablation studies discussed in the main paper, validating the efficacy of PRISM's causal design, the selection of physics priors, and the supervision strategy. Furthermore, we conduct a sensitivity analysis on key hyperparameters to demonstrate the robustness of our method.

### D.1. Detailed Ablation Studies

*1) Impact of Causal Architecture.* We validate the "Reason $\rightarrow$ Disentangle $\rightarrow$ Generate" paradigm by selectively removing components from the PRISM pipeline. The quantitative results are summarized in Table 15.

- **NRG Only (Baseline):** Utilizing a vanilla diffusion model without physics priors results in poor performance (Inv-IoU 0.412), as the model fails to distinguish between aleatoric sensor noise and deterministic physical invalidations.

- **w/o SPR (Physics-Blind):** Removing the VFM-driven semantic priors leads to a significant drop in invalidation detection (F1 0.891 vs. 0.975). This confirms that without semantic reasoning, the model struggles to infer material-dependent failures, such as those caused by non-Lambertian surfaces.

- **w/o BND (Monolithic):** Bypassing the disentangler forces the NRG to learn a conflated noise distribution. This results in the highest measurement error (MAE 0.098m vs. 0.076m) and severe geometric hallucinations, proving that explicit disentanglement is crucial for preventing the "averaging effect" in noise synthesis.

*2) Efficacy of Physics Priors (VFM Backbones).* Table 16 compares generic vision backbones against geometry-aware 3D VFMs. Results indicate that 3D VFMs (e.g., MoGe2, Depth Anything V2) significantly outperform generic ViTs (DINOv2) and CNNs. Notably, MoGe2-Large achieves the highest Robot SR (92.5%), suggesting that pre-trained geometric foundation models implicitly encode robust semantics-physics correlations (e.g., material $\leftrightarrow$ sensor failure pattern), which are more transferable to sensor simulation than purely semantic features. While larger models incur higher inference latency, the performance gain justifies the cost for offline data generation.

*3) Analysis of Supervision Strategy (H-PPS).* We dissect the contributions of each component in our Hierarchical Positive-Prioritized Supervision (H-PPS) in Table 17.

*Table 15.* **Component Ablation Study.** We evaluate the contribution of Semantic-Physics Reasoner (SPR) and Bimodal Noise Disentangler (BND) against the pure Noise Residual Generator (NRG) baseline. **Inv-IoU** and **Inv-F1** measure the detection of sensor holes, while **Acc-MAE** measures the precision in valid regions. **Robot SR** denotes the success rate in Sim2Real manipulation.

| Method | Paradigm *Config* | Overall Metrics | | Invalidation (Safety) | | Accuracy (Precision) | | Downstream |
|---|---|---|---|---|---|---|---|---|
| | | MAE ↓ | $\delta < 1.25$ ↑ | IoU ↑ | F1 ↑ | MAE ↓ | $\delta < 1.05$ ↑ | Robot SR ↑ |
| **NRG Only** | *Baseline* | 0.118 | 87.3% | 0.412 | 0.583 | 0.089 | 71.5% | 72.5% |
| **w/o SPR** | *Physics-Blind* | 0.095 | 91.8% | 0.823 | 0.891 | 0.068 | 82.4% | 83.7% |
| **w/o BND** | *Monolithic* | 0.098 | 90.6% | 0.756 | 0.852 | 0.071 | 80.8% | 81.5% |
| **PRISM (Full)** | *Ours* | **0.076** | **96.4%** | **0.952** | **0.975** | **0.052** | **91.3%** | **92.5%** |

*Table 16.* **VFM Backbone Comparison in SPR.** We compare CNNs, Generic ViTs, and 3D VFMs. **Robot SR** refers to the average success rate across all 6 manipulation tasks. **Inference** time is measured on a single H100 GPU.

| Backbone | Type | Depth Fidelity | | Physics Awareness | | Efficiency | |
|---|---|---|---|---|---|---|---|
| | | MAE ↓ | $\delta < 1.25$ ↑ | Inv-IoU ↑ | Robot SR ↑ | Params | Inference |
| ResNet-50 (Scratch) | CNN | 0.142 | 82.6% | 0.685 | 60.0% | 25M | **12ms** |
| ResNet-50 (ImageNet) | CNN | 0.125 | 86.1% | 0.738 | 67.0% | 25M | 12ms |
| DINOv2-Base | Generic ViT | 0.108 | 89.7% | 0.812 | 73.5% | 86M | 28ms |
| DINOv2-Large | Generic ViT | 0.095 | 92.3% | 0.856 | 78.0% | 304M | 45ms |
| DepthAnything-V2-L | 3D VFM | 0.082 | 95.1% | 0.925 | 88.0% | 335M | 48ms |
| **MoGe2-Large (Ours)** | 3D VFM | **0.076** | **96.4%** | **0.952** | **92.5%** | 350M | 52ms |

- **Class Imbalance:** Standard BCE yields a poor recall of 48.5% due to the sparsity of invalidation regions. Introducing the positive weight ($w_{pos} = 3.0$) drastically boosts recall to 71.2%.

- **Shape & Boundary:** Integrating Dice Loss improves global shape compactness (IoU +8.3%), while the proposed PP-DHM is critical for fine-grained details. Compared to static mining, our dynamic decay strategy $\gamma(t)$ improves **Boundary-IoU** by 8.8%, ensuring sharp delineation of sensor artifacts.

*Table 17.* **Progressive Ablation of H-PPS.** We add components sequentially to the standard BCE baseline. **Boundary-IoU** evaluates the precision of artifact edges (within 5px).

| Config | Mechanism Added | Invalidation Detection | | | Fine-Grained |
|---|---|---|---|---|---|
| | | IoU ↑ | F1 ↑ | Recall ↑ | Boundary-IoU ↑ |
| **Baseline** | BCE Only | 0.548 | 0.642 | 0.485 | 0.312 |
| + Weight | Positive-Weighted ($w_{pos} = 3.0$) | 0.685 | 0.756 | 0.712 | 0.425 |
| + Dice | Shape Constraint | 0.768 | 0.845 | 0.792 | 0.538 |
| + Aux | Multi-Scale Supervision | 0.856 | 0.912 | 0.875 | 0.652 |
| + Mining | PP-DHM (Static $\gamma = 0.25$) | 0.912 | 0.948 | 0.945 | 0.735 |
| **H-PPS (Ours)** | **Dynamic Decay $\gamma(t)$** | **0.952** | **0.975** | **0.991** | **0.823** |

## D.2. Hyperparameter Sensitivity

To demonstrate the robustness of PRISM and justify our design choices, we analyze the sensitivity of the supervision strategy to the positive weight $w_{pos}$ and the dynamic mining schedule.

**1) Impact of Positive Weight $w_{pos}$.** The positive weight balances precision and recall for sparse invalidation masks. As shown in Table 18 (Left), setting $w_{pos} = 1.0$ (standard BCE) results in a dangerously low recall (0.485), causing the robot to miss potential collision risks (blind spots). Increasing $w_{pos}$ improves recall significantly. We observe that $w_{pos} = 3.0$ achieves the optimal trade-off with the highest F1-score (0.975). Further increasing $w_{pos}$ to 5.0 yields diminishing returns in

recall (0.995) but significantly degrades precision (0.812) and IoU, as the model becomes overly aggressive and hallucinates invalidations in clean regions.

**2) Impact of Dynamic Mining Strategy** ($\gamma_{start} \to \gamma_{end}$)**.** We employ a dynamic decay strategy for the hard mining ratio, decaying from $\gamma_{start}$ to $\gamma_{end}$. We fix $\gamma_{end} = 0.1$ to ensure aggressive boundary refinement in the final stages and analyze the impact of the initial ratio $\gamma_{start}$ in Table 18 (Right). **Too Small** ($\gamma_{start} = 0.10$)**:** Contrary to the intuition that "harder is better," starting with a very small ratio degrades performance (IoU 0.905). In the early training phase, focusing on an extremely small subset of pixels causes the model to *lose global optimization direction*, leading to instability and slower convergence. **Too Large** ($\gamma_{start} = 1.00$)**:** Starting without mining includes too many easy negatives, diluting the gradient signal for boundary learning. **Optimal** ($\gamma_{start} = 0.25$)**:** We find that starting at 0.25 provides a "warm start" with sufficient global context while focusing on difficult regions. Combined with the decay to $\gamma_{end} = 0.1$, this "Start-Large-End-Small" strategy yields the best Boundary-IoU (0.823).

*Table 18.* **Sensitivity Analysis.** We analyze the trade-offs for Positive Weight $w_{pos}$ (Left) and Initial Mining Ratio $\gamma_{start}$ (Right). The optimal configurations ($w_{pos} = 3.0$ and $\gamma_{start} = 0.25$) represent the best balance.

| | **Impact of Positive Weight** $w_{pos}$ | | | **Impact of Mining Ratio** $\gamma_{start}$ | | |
|---|---|---|---|---|---|---|
| $w_{pos}$ | Recall ↑ | Precision ↑ | F1 ↑ | $\gamma_{start}$ | Inv-IoU ↑ | B-IoU ↑ |
| 1.0 | 0.485 | **0.962** | 0.642 | 0.10 | 0.905 | 0.752 |
| 2.0 | 0.824 | 0.951 | 0.883 | **0.25** | **0.952** | **0.823** |
| **3.0** | **0.991** | 0.959 | **0.975** | 0.50 | 0.931 | 0.789 |
| 4.0 | 0.993 | 0.885 | 0.936 | 0.75 | 0.884 | 0.741 |
| 5.0 | 0.995 | 0.812 | 0.894 | 1.00 | 0.856 | 0.705 |

# E. Additional Qualitative Visualizations

In this section, we provide an extensive gallery of synthesized results to visually validate the fidelity of PRISM. We also present frame-by-frame rollouts of robotic manipulation to demonstrate the practical impact of our method.

### E.1. Depth Synthesis Gallery

**Diverse Material Scenarios.** Fig. 10 to Fig. 14 illustrate synthesized depth maps across a wide spectrum of material properties, ranging from highly specular metals to semi-transparent plastics. Please refer to the captions for detailed comparisons against baselines.

**Failure Case Analysis.** While PRISM achieves state-of-the-art performance, it is not without failure modes. We identify two primary failure scenarios:**Semantic Ambiguity in VFMs:** In rare cases where the frozen VFM misinterprets the scene semantics (e.g., mistaking a mirror for a window due to reflection), the SPR module generates incorrect physics priors, leading to erroneous invalidation masks. **Ultra-Thin Structures:** For objects with geometry thinner than the VFM's patch resolution (e.g., fine wires < 2mm), the BND module may over-smooth the invalidation boundaries, failing to capture the "flying pixel" effect accurately.

### E.2. Robot Manipulation Rollouts

To demonstrate why PRISM-trained policies succeed where others fail, we visualize the decision-making process in dynamic environments. Fig. 15 to Fig. 20 presents time-synchronized snapshots rollout of all tasks. The PRISM-trained policy correctly identifies the noisy transparent surface as an obstacle and reorients the gripper to grasp the opaque handle, demonstrating robust geometric reasoning under uncertainty.

# F. Limitations and Future Work

While PRISM establishes a rigorous, physics-grounded paradigm for sensor modeling, its current scope reveals several strategic opportunities for future exploration. We view these not merely as boundaries of our current implementation, but as **critical gateways** to the next generation of trustworthy embodied AI.

**1. Paradigm Shift towards Noise-Native Policy Learning.** Unlike online domain adaptation methods that introduce inference latency, PRISM operates as an offline data generator, fundamentally solving the "realism vs. efficiency" trade-off. This opens a high-value research direction: **shifting the focus from online denoising to intrinsic policy robustness.** Since PRISM can synthesize theoretically infinite, physically valid noisy data, future work can abandon auxiliary depth refinement modules during deployment. Instead, researchers can leverage PRISM to design "noise-native" policies—architectures trained with "curriculum noise" that treat sensor artifacts not as errors to be removed, but as features to be understood. This could significantly simplify the deployment stack and enhance robustness against adversarial sensor attacks.

**2. Cross-Sensor Generalization.** PRISM's SPR and BND modules are trained on a fixed sensor family (Active Stereo), and their learned semantic-to-noise mappings do not transfer reliably to sensors with fundamentally different noise formation principles, such as Time-of-Flight (ToF) or LiDAR. Specifically, ToF sensors exhibit noise patterns driven by multipath interference and reflectivity rather than stereo matching failure, while LiDAR returns sparse but highly accurate measurements. A promising direction is to develop lightweight per-sensor adaptation modules—such as low-rank prompt layers appended to SPR's attention outputs—that re-calibrate the semantic-to-failure mapping from a small number of target-sensor samples, enabling few-shot cross-sensor transfer without full retraining.

**3. Temporal Consistency in Dynamic Scenes.** PRISM currently models sensor noise on a per-frame basis. However, real-world sensor noise often exhibits temporal correlations (e.g., flickering invalidation masks on static surfaces caused by speckle instability or rolling-shutter motion). In highly dynamic scenes, independent per-frame inference ignores these temporal structure, causing invalidation predictions to flicker inconsistently across frames even for physically stable surfaces. Extending PRISM with a flow-guided temporal state module could propagate noise state across frames, enforcing temporal coherence in the synthesized invalidation masks and residuals, which is crucial for policies that rely on multi-frame geometric reasoning.

**4. Interaction-Induced Extrinsic Noise.** Our current formulation focuses on *intrinsic* sensor noise correlated with surface appearance and scene geometry. It does not explicitly model *extrinsic* noise sources arising from the robot's physical interaction with the environment, such as camera vibrations (sensor shake) or multi-camera interference (crosstalk) in multi-robot setups. Integrating these extrinsic, interaction-coupled factors into the causal graph would be a valuable extension for complex, multi-agent robotic systems.

**5. Adaptation to Extreme Optical Environments.** Our current SPR module relies on general-purpose VFMs (e.g., MoGe) trained primarily on terrestrial, indoor imagery. Consequently, PRISM's performance may degrade in domains where the dominant noise formation mechanism diverges from surface-reflectance-driven failure—such as **dense fog, rain, or sandstorms** in outdoor autonomous driving scenarios. In these environments, scattering and atmospheric attenuation dominate over material-surface interactions. A promising future direction is to develop lightweight domain-specific adaptation layers that re-calibrate VFM priors toward environment-dominant noise statistics, extending PRISM's sensor modeling applicability beyond controlled indoor manipulation to all-weather deployment scenarios.

**Conclusion.** Despite these limitations, PRISM demonstrates that explicitly disentangling physical causality from semantic priors is key to bridging the perceptual Sim-to-Real gap. We hope this work inspires the community to move beyond monolithic noise modeling toward interpretable, physics-grounded generative simulation.

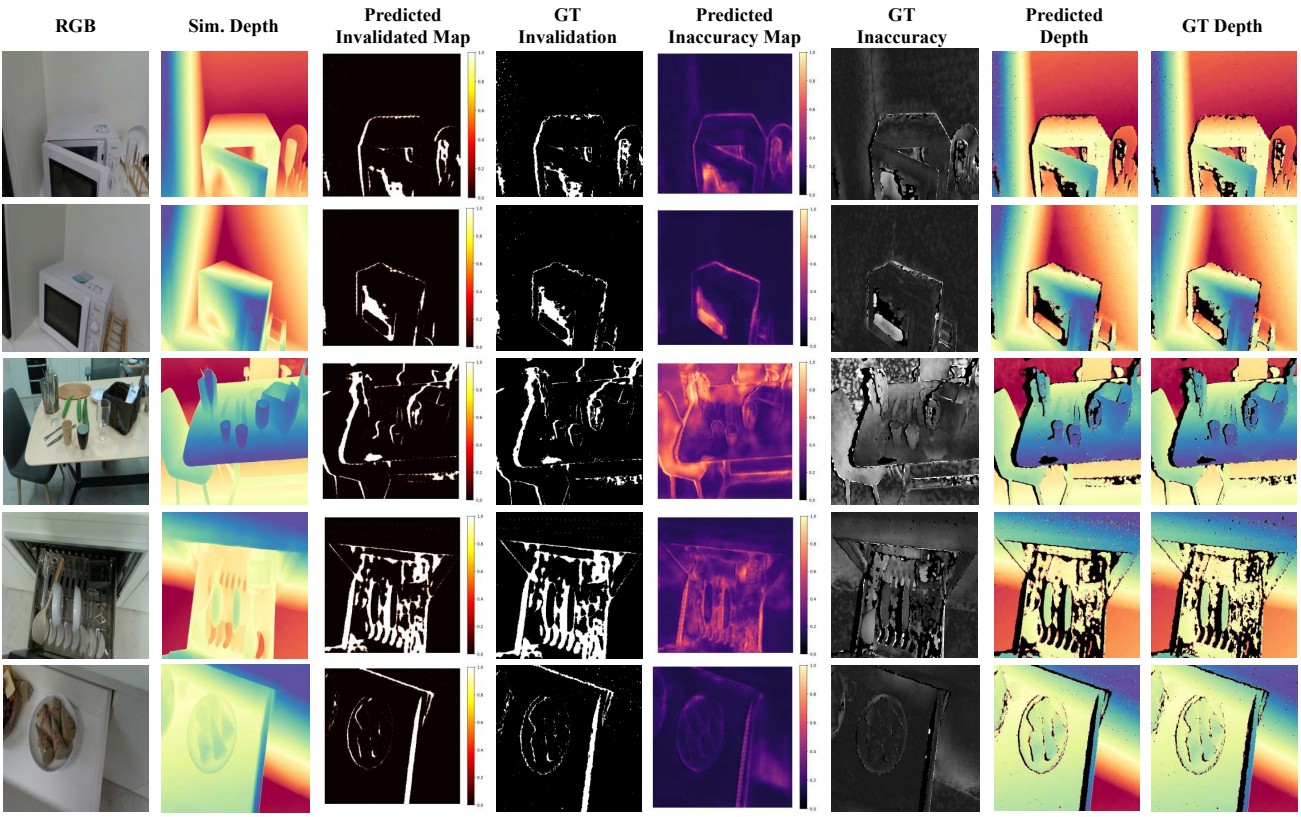

*Figure 10.* **Qualitative synthesis on the *Kitchen* scene**.

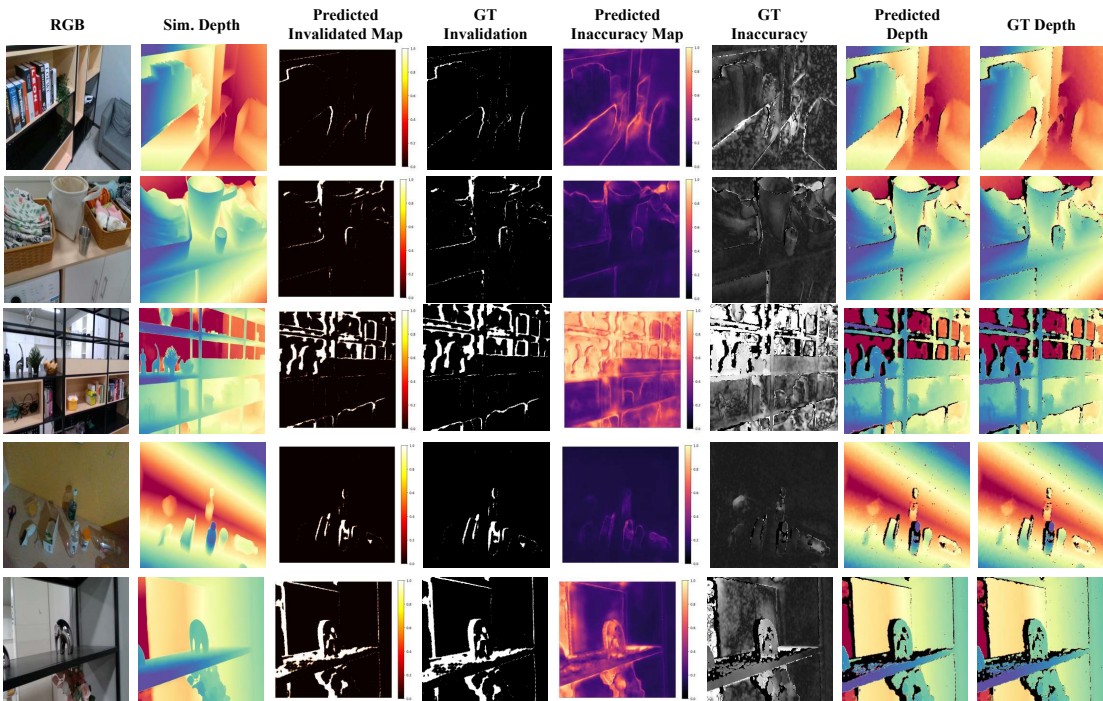

*Figure 11.* **Qualitative synthesis results on the *Living Room* scene**.

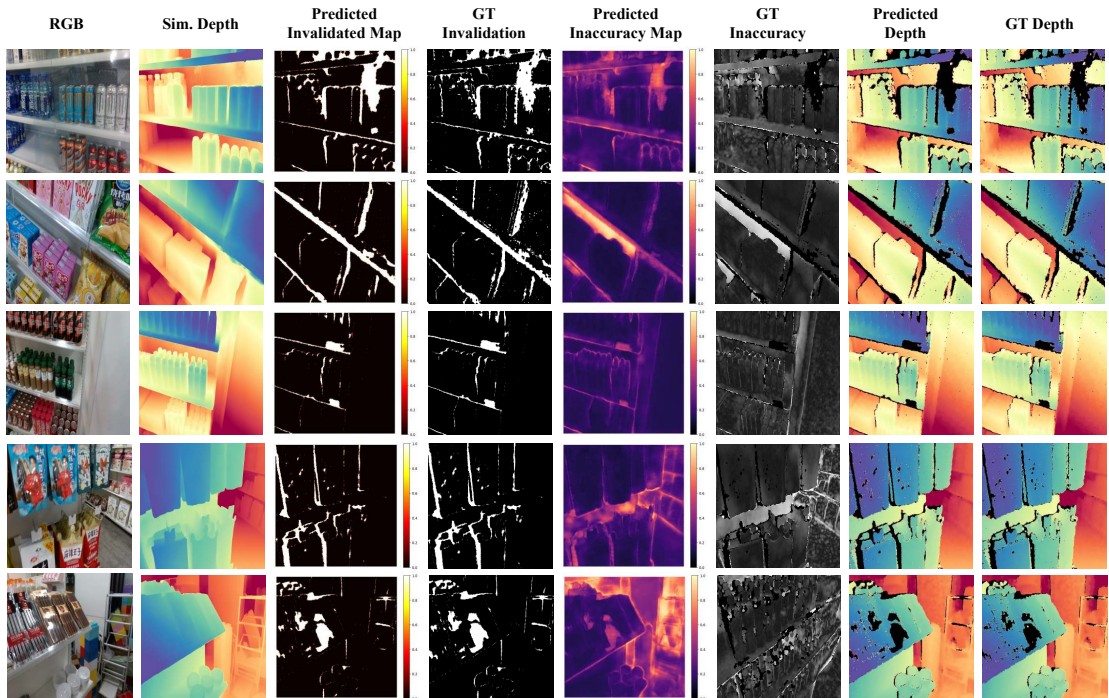

*Figure 12.* **Qualitative synthesis results on the *Market* scene**.

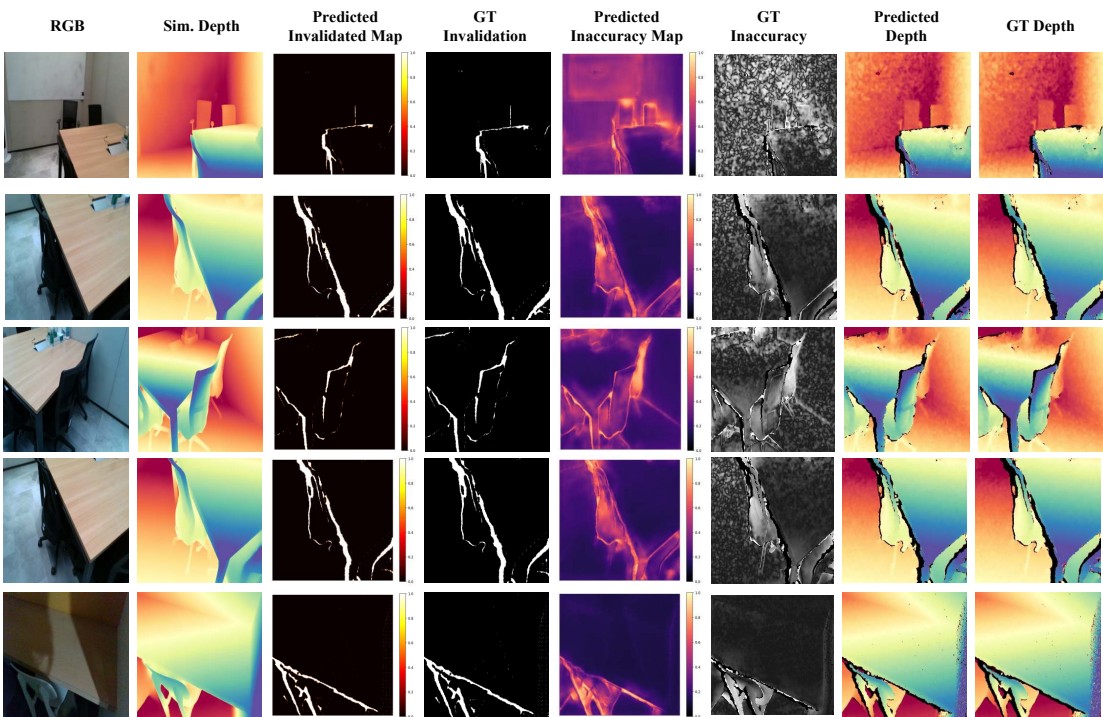

*Figure 13.* **Qualitative synthesis results on the *Office* scene**.

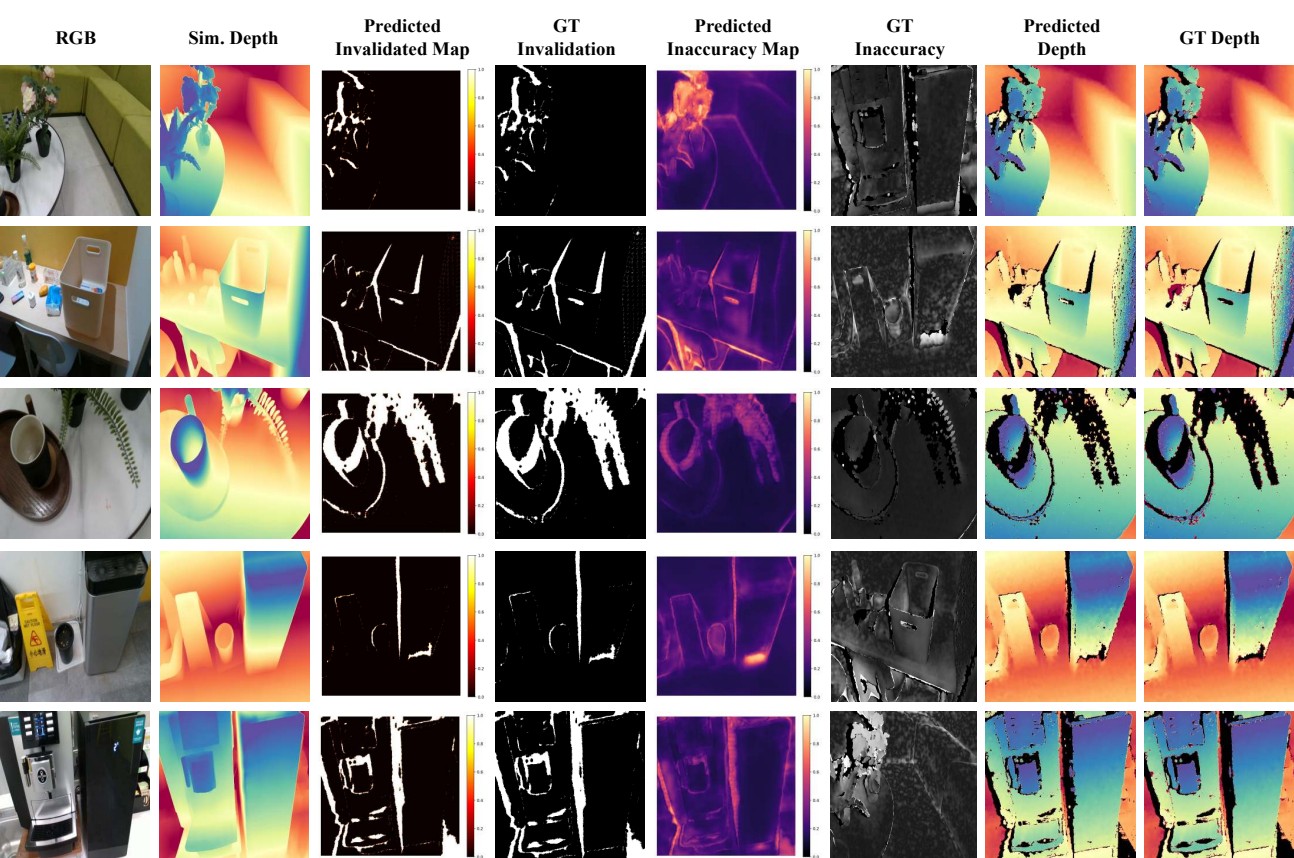

*Figure 14.* **Qualitative synthesis results on the *Lounge* scene**.

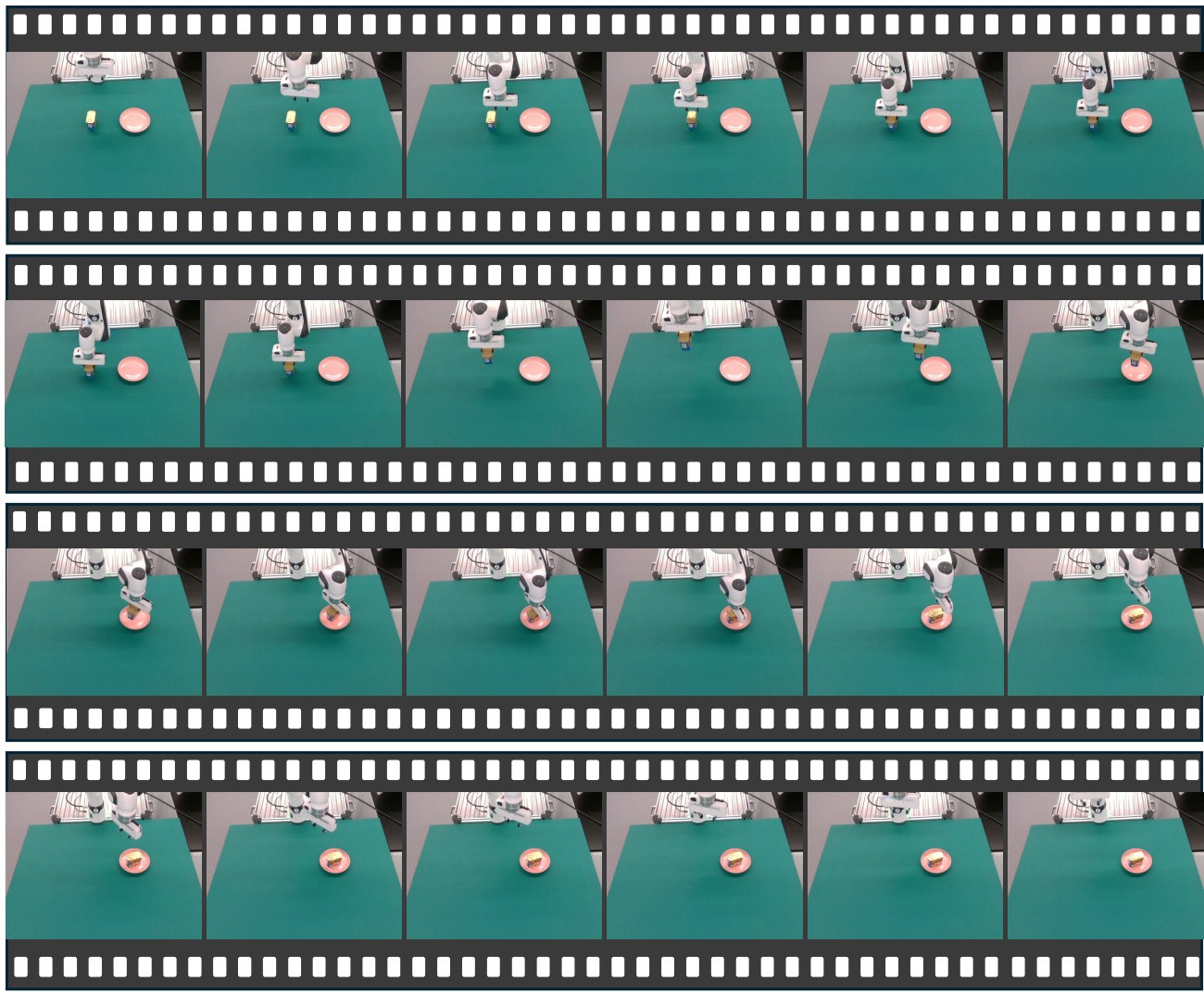

*Figure 15.* **Real-world rollout for T1 (Can-to-Plate)**.

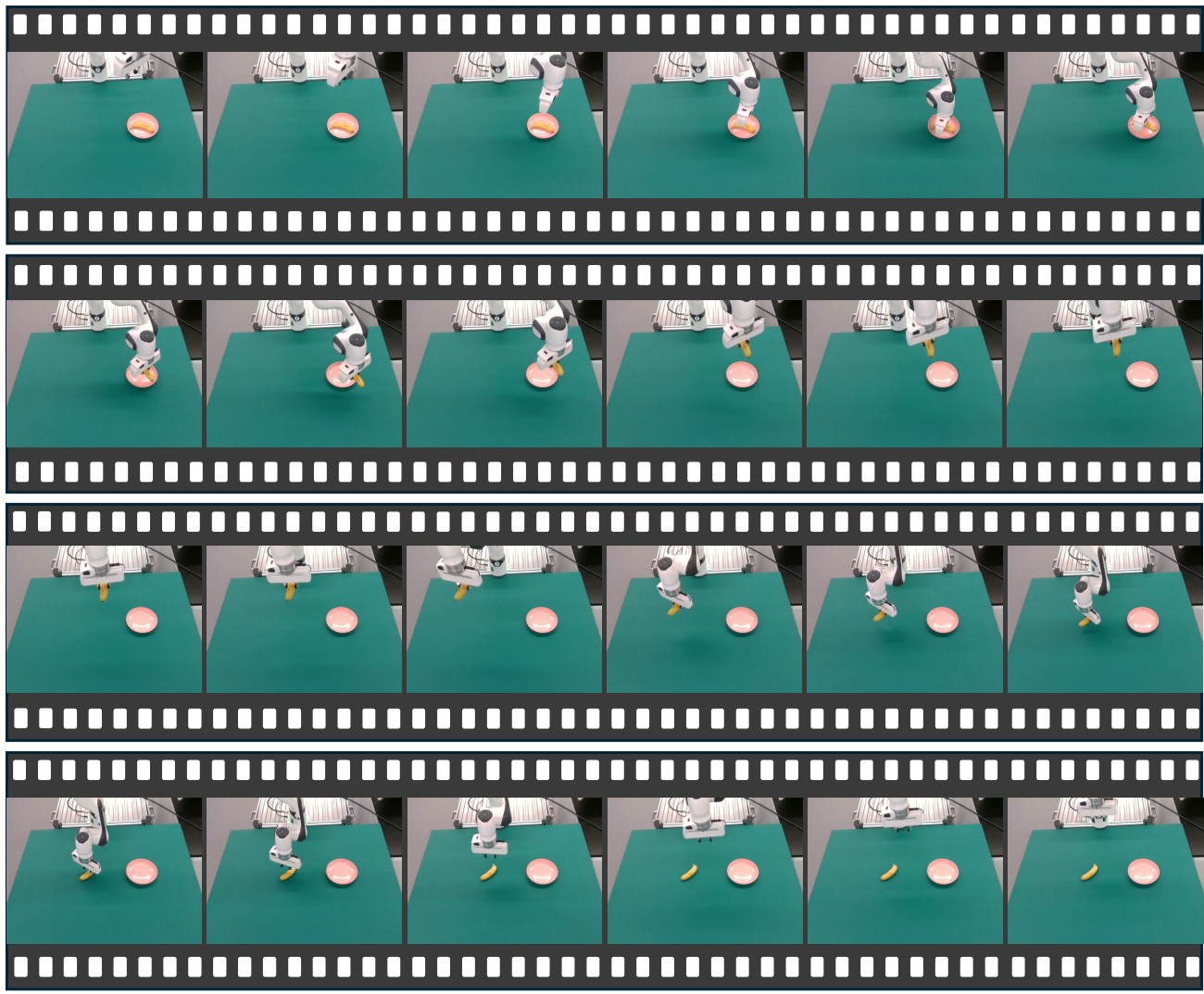

*Figure 16.* **Real-world rollout for T2 (Banana-to-Plate)**.

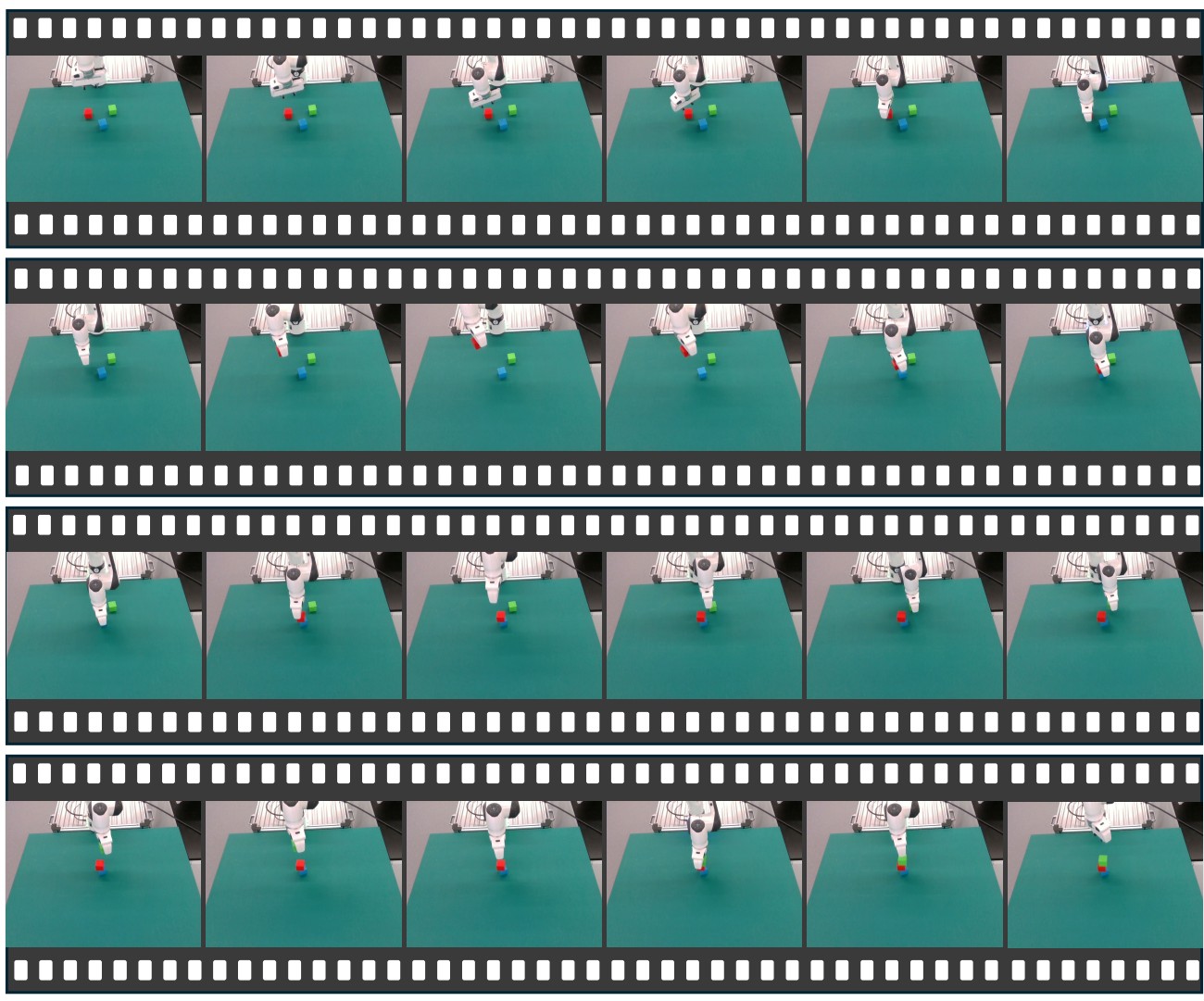

*Figure 17.* **Real-world rollout for T3 (Stack Cubes)**.

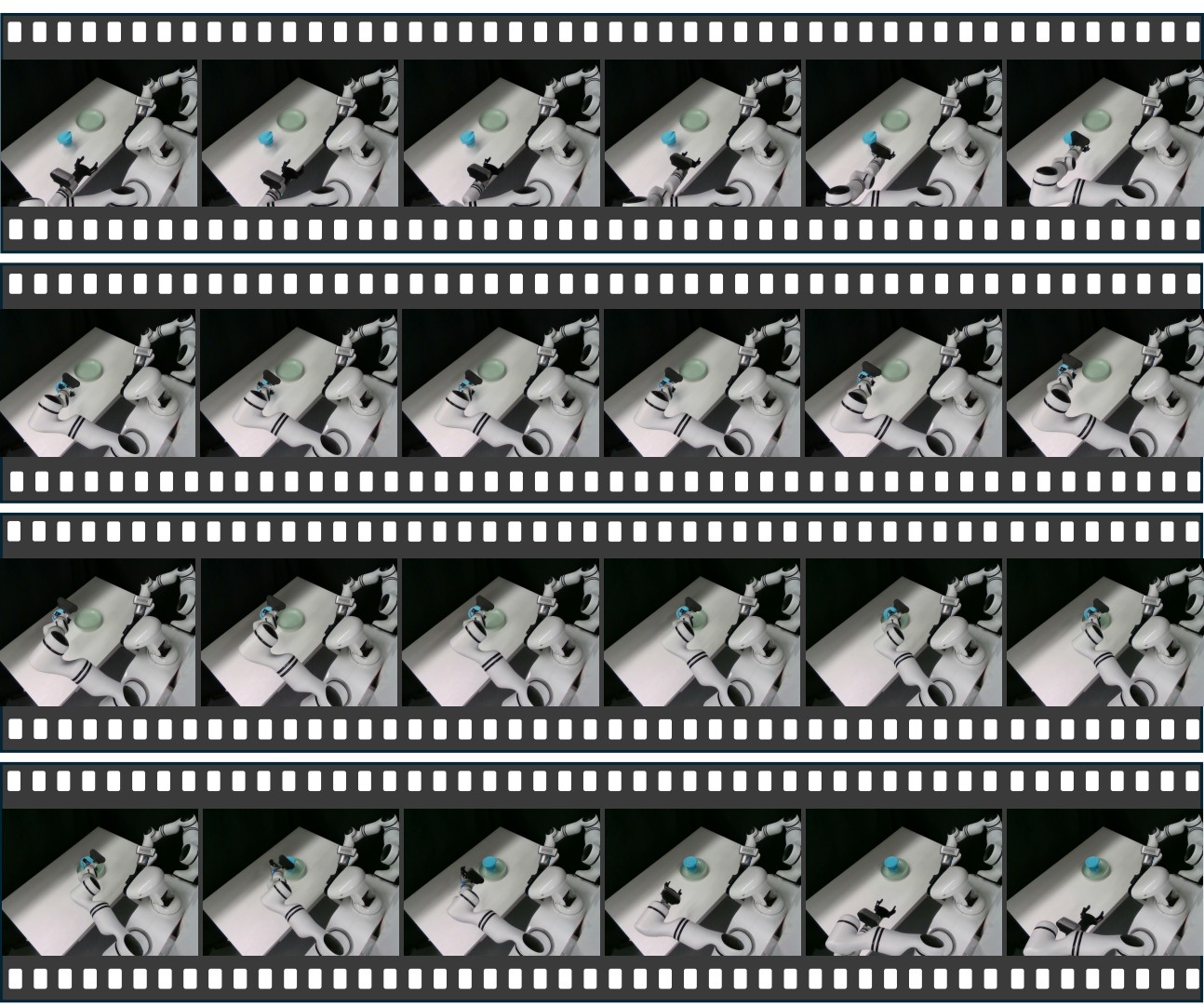

*Figure 18.* **Real-world rollout for T4 (Bucket-to-Plate)**.

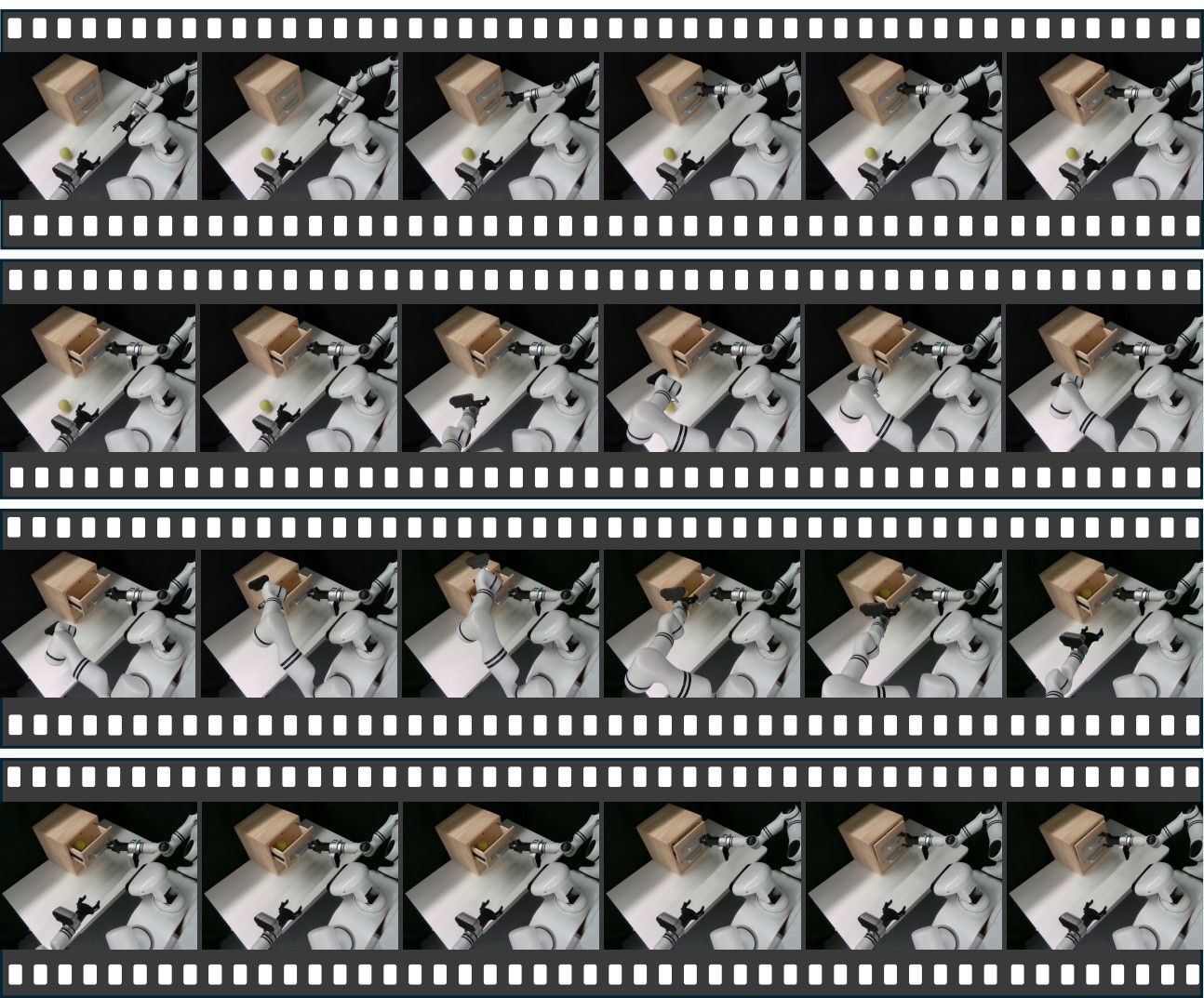

*Figure 19.* **Real-world rollout for T5 (Toyball-to-Drawer)**.

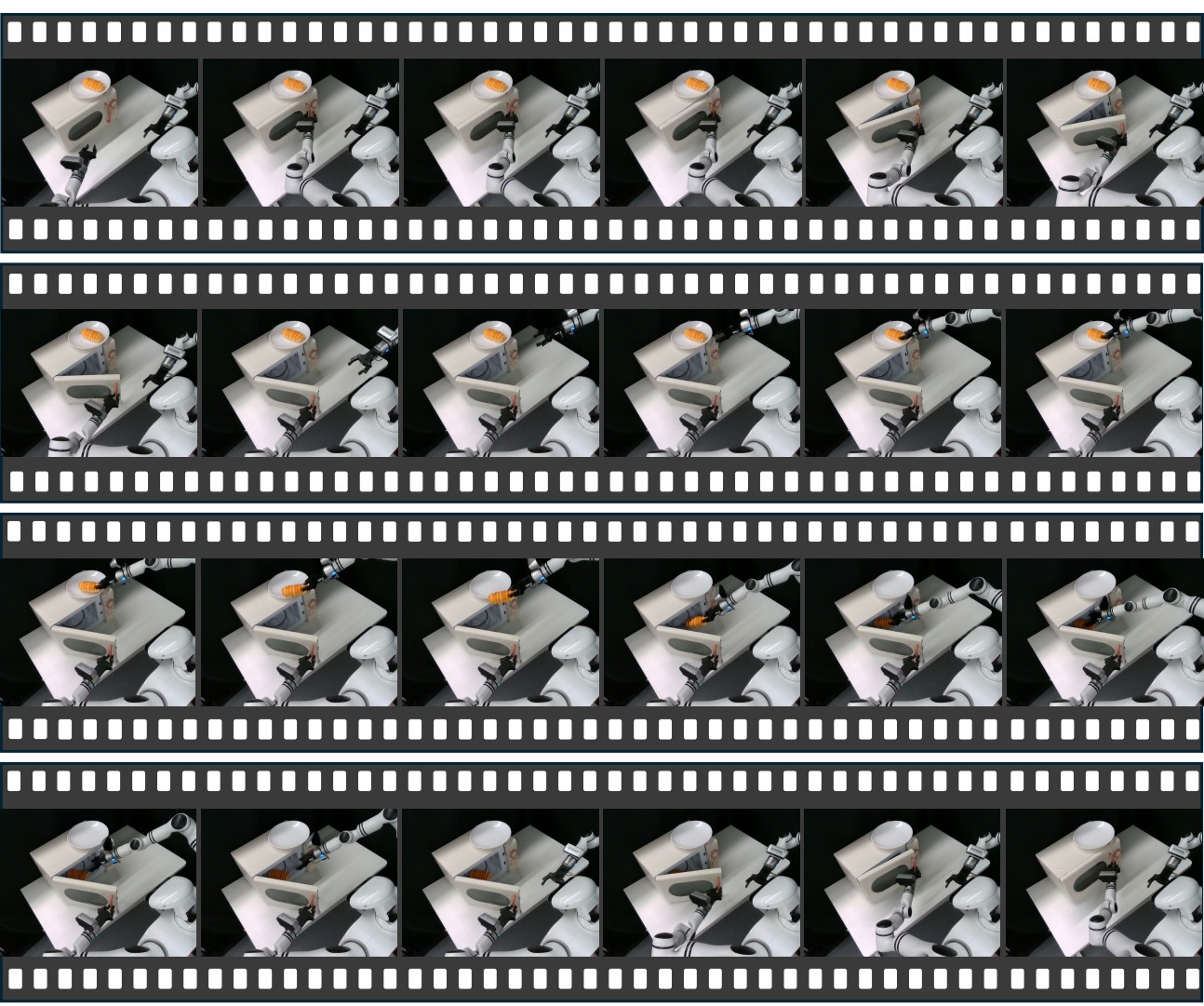

*Figure 20.* **Real-world rollout for T6 (Bread-to-Microwave)**.

