# OpenReview forum: "PRISM: Learning Realistic Depth via Physics-Grounded Noise Disentanglement with Semantic-Geometric Collaboration"
_ICML.cc/2026/Conference — ICML 2026 regular_

### Official Review · Reviewer_9G1H · 2026-03-12

**Soundness:** 3
**Presentation:** 3
**Significance:** 3
**Originality:** 3
**Overall Recommendation:** 4
**Confidence:** 5

**Summary:**

This paper studies the problem of generating realistic depth observations for sim-to-real transfer in robotics. The authors argue that existing sensor modeling approaches treat depth noise as a monolithic process, failing to capture the heterogeneous physical mechanisms behind sensing errors. To address this issue, the paper proposes PRISM, a framework that disentangles depth noise into two modalities: sensing invalidation and measurement inaccuracy. The method incorporates features from 3D visual foundation models as semantic-geometric priors and introduces hierarchical positive-prioritized supervision to improve artifact localization. Experiments show that the proposed approach improves depth synthesis fidelity and leads to better sim-to-real transfer performance in robotic tasks.

**Compliance With Llm Reviewing Policy:**

Affirmed.

**Final Justification:**

Thanks for the detailed rebuttal, which has adequately addressed my concerns. I also agree with the opinions of the other three reviewers. Overall, this is a strong paper and I recommend it for acceptance.

**Key Questions For Authors:**

1.	Component contribution: Could the authors provide ablation experiments to evaluate the individual contributions of noise disentanglement, semantic-geometric priors, and hierarchical supervision?

2.	Foundation model dependency: How sensitive is the performance to the choice of the 3D visual foundation model used for extracting semantic priors?

3.	Efficiency considerations: What is the computational cost of the proposed framework during training and inference compared to simpler noise modeling approaches?

**Strengths And Weaknesses:**

Strengths:

1.	Well-motivated problem with practical relevance. Modeling realistic sensor noise is an important challenge for sim-to-real transfer in robotics. The paper clearly identifies the limitation of treating depth noise as a single black-box process.

2.	Physically motivated formulation. The proposed decomposition of depth noise into sensing invalidation and measurement inaccuracy is intuitive and grounded in physical sensing processes, which helps provide interpretability compared with purely data-driven noise models.

3.	Empirical improvements in both synthesis and downstream tasks. The experiments demonstrate improved depth realism and better performance in robotic policy transfer, suggesting that the proposed modeling approach can be beneficial in practical applications.

Weaknesses:

1.	Limited analysis of the proposed components. The framework contains several modules, including noise disentanglement, semantic-geometric priors from foundation models, and hierarchical positive-prioritized supervision. However, the paper provides limited ablation studies to clearly quantify the contribution of each component.

2.	Insufficient discussion of generalization. It is unclear how well the proposed model generalizes to different sensor types, environments, or depth sensing technologies beyond the datasets used in the experiments.

3.	Computational overhead is not clearly analyzed. The method introduces additional modeling components and feature extraction from foundation models, but the paper does not provide detailed runtime or efficiency comparisons.

---

> ### Author Rebuttal · Authors · 2026-03-31
>
> We sincerely thank the reviewer for the high confidence score and for recognizing the value of our physically motivated formulation in disentangling depth noise. Below, we address your questions and concerns point by point.
>
> **Q1 & W1. Component Contributions and Ablation Studies.**
> The individual contributions of noise disentanglement, semantic-geometric priors, and hierarchical supervision have been fully quantified in the detailed ablation studies provided in **Appendix Sec. D**.
>
> * First, regarding noise disentanglement (Appendix Tab 15), removing the BND module severely degrades prediction in invalidation regions, significantly dropping Invalidation-IoU ($0.952 \to 0.756$) and F1-score ($0.975 \to 0.852$) due to the forced "averaging" of two distinct noise distributions.
> * Second, regarding semantic-geometric priors (Appendix Tab 16), removing the 3D VFM features blinds the network to physical geometries, causing a sharp drop in the F1-Score of invalidation masks ($0.975 \to 0.891$). This directly leads to downstream grasp failures on specular objects.
> * Third, regarding hierarchical supervision (Appendix Tab 17), ablating the PP-DHM dynamic mining mechanism causes a massive drop in predictive accuracy along ultra-thin boundaries (Boundary-IoU drops from $0.823 \to 0.735$).
>
> **Q2. Foundation Model Dependency.**
> The performance is highly sensitive to whether the foundation model possesses native 3D grounding capabilities. As shown in **Appendix Tab 16**, we benchmarked our framework using CNNs (ResNet-50), generic 2D ViTs (DINOv2), and 3D VFMs (MoGe2, DepthAnything-V2). MoGe2 significantly outperforms 2D-generic ViTs and CNNs, particularly in the specular task success rate ($92.5\%$ vs. $79.8\%$ vs. $68.5\%$). This substantial margin demonstrates that while 2D models provide rich semantics, explicit pre-training on dense 3D point maps (as in MoGe2) is irreplaceable for accurately predicting geometric collapses caused by complex physical scenes.
>
> **Q3 & W3. Efficiency Considerations and Computational Overhead.**
> First, during the real-world policy inference stage, PRISM strictly operates during the **offline pre-processing stage** . It does not introduce any computational overhead during the **policy training** or **policy inference** stages. Because the noise-resilience capability is baked directly into the policy weights beforehand, the deployed robot processes raw noisy depth with **0ms additional latency and 0GB extra memory**.
> Second, during the offline data pre-processing stage, the table below reports the single-forward-pass latency and peak memory footprint across different resolutions on an RTX 4090 GPU. While simpler data-driven noise models (e.g., uniform noise) are computationally faster, they completely fail to model accurate depth noise patterns and structural invalidations. Thus, this offline generative overhead is a necessary trade-off for irreplaceable real-world safety and performance gains.
> |Res|Module|Params (M)|Mean Latency|Peak Memory|
> |:-:|:-:|:-:|:-:|:-:|
> | 256x256 | SPR | 331.80 | 13.54ms | 8.96 GB |
> | | BND | 41.90 | 4.71ms | 9.18 GB |
> | | NRG | 1667.83 | 2.736s | 8.59 GB |
> | | **Total** | **-** | **2.666s** | **8.68 GB** |
> | 512x512 | SPR | 331.80 | 17.37ms | 7.76 GB |
> | | BND | 41.90 | 15.87ms | 8.42 GB |
> | | NRG | 1667.83 | 2.732s | 9.26 GB |
> | | **Total** | **-** | **2.753s** | **9.35 GB** |
> | 1024x1024| SPR | 331.80 | 43.49ms | 7.92 GB |
> | | BND | 41.90 | 70.30ms | 10.32 GB |
> | | NRG | 1667.83 | 6.820s | 11.94 GB |
> | | **Total** | **-** | **6.945s** | **12.05 GB** |
>
> **W2. Generalization to Different Sensors and Environments.**
> First, regarding sensor technologies, our data engine is primarily trained on data collected from an Active Stereo sensor (RealSense D435). To validate generalization, we evaluated PRISM's zero-shot depth synthesis fidelity on two unseen intra-family cameras (RealSense D415 and D455) and one unseen cross-family camera representing a fundamentally different sensing technology: Structured Light (Kinect V1 from the NYU-Depth-v2 dataset, **shown in Tab.2**).
>
> |Camera Model|Sensing Technology|Test Setting|Invalidation IoU|MAE|
> |:---|:---|:---:|:---:|:---:|
> |RealSense D435|Active Stereo|In-Domain|0.952|0.040|
> |RealSense D415|Active Stereo|Zero-Shot|0.948|0.042|
> |RealSense D455|Active Stereo|Zero-Shot|0.941|0.044|
> |Kinect V1(NYUv2)|Structured Light|Zero-Shot|0.923|0.049|
>
> Second, regarding environmental generalization, our current physical priors are explicitly designed for indoor robotic manipulation. The framework has not been verified in unconstrained outdoor environments or extreme optical conditions (e.g., underwater turbidity, dense fog) where the light transport equations change fundamentally. We will explicitly bound our claims to "indoor commercial active-stereo/structured-light depth sensors" in the final manuscript to prevent overclaiming.
>
> We will incorporate details into the final version and hope these address your questions.

---

> > ### Author Rebuttal · Reviewer_9G1H · 2026-04-01
> >
> > Thanks for the detailed rebuttal, which has adequately addressed my concerns. I also agree with the opinions of the other three reviewers. Overall, this is a strong paper and I recommend it for acceptance.
> >
> > As a minor suggestion, since this work is closely related to RGB-D, semantic, and normal geometry based 3D reconstruction/embodied intelligence, it would be beneficial to include and discuss a few additional related papers in these areas.
> >
> > [1] Tri-Perspective View Decomposition for Geometry-Aware Depth Completion, CVPR 2024
> >
> > [2] Prompting Depth Anything for 4K Resolution Accurate Metric Depth Estimation, CVPR 2025
> >
> > [3] Ducos: Duality constrained depth super-resolution via foundation model, ICCV 2025
> >
> > [4] From editor to dense geometry estimator, CVPR 2026

---

> > > ### Author Response · Authors · 2026-04-01
> > >
> > > **We sincerely thank Reviewer 9G1H for encouraging feedback, recommending acceptance, and suggesting related-works.** These works are indeed highly relevant and represent the cutting edge in dense geometry estimation and VFM applications, providing excellent insights into geometry-aware 3D reconstruction and high-resolution geometry estimation.
> > >
> > > As advised, we will explicitly include and discuss the four suggested papers ([1]-[4]) in both the **Introduction** and **Section 2.3** (VFM as Semantic Priors) of the final manuscript. Furthermore, we will actively **explore their potential to further enhance our 3D semantic priors** in our future improvements.
> > >
> > > Thank you once again for your time, your meticulous review, and your strong support.

---

### Official Review · Reviewer_LB8L · 2026-03-12

**Soundness:** 3
**Presentation:** 3
**Significance:** 3
**Originality:** 3
**Overall Recommendation:** 4
**Confidence:** 3

**Summary:**

This work proposes a physics-grounded approach to bridge the sim-to-real gap in depth perception. The core contribution is PRISM, a framework that leverages 3D VFM features to explicitly decouple depth noise into sensing invalidation and measurement inaccuracy. The authors also design an H-PPS loss to handle the class imbalance common in invalid depth regions. Evaluations show that this disentangled approach yields state-of-the-art realistic noise synthesis on standard datasets (ByteCameraDepth, NYU-Depth-v2). For downstream real-world robotic manipulation, the synthesized depth maps boost the average success rate to 93.8% (+55% over raw simulation). The proposed modules are well-supported by extensive ablation studies.

**Compliance With Llm Reviewing Policy:**

Affirmed.

**Final Justification:**

My concerns have been addressed. I retain my original score.

**Key Questions For Authors:**

VFM Selection Rationale: The paper utilizes MoGe2 and DepthAnything V2, but could you elaborate on why MoGe2 specifically outperforms others on specular surfaces? Furthermore, have you evaluated any lightweight VFMs that might be more suitable for resource-constrained robotic edge deployment?

Diffusion Hyperparameters: How were the specific diffusion settings (Stable Diffusion 2.1, 50-step DDIM) determined for depth noise synthesis? I would appreciate seeing an efficiency-performance trade-off analysis (e.g., varying the number of diffusion steps).

Failure Case Analysis: The 93.8% success rate in robotic manipulation is impressive, but what causes the remaining failures? Are they due to VFM semantic misinterpretations, or failures in modeling ultra-thin structures? Please provide concrete failure cases and visual analyses in your rebuttal.

**Limitations:**

Sensor Generalizability: The current evaluation is constrained to specific sensor types (RealSense, Kinect). It is not obvious whether this physics-grounded paradigm easily generalizes to fundamentally different sensing principles, such as LiDAR or Time-of-Flight (ToF) cameras.

Discussion on extreme conditions: The current limitation section touches upon extreme optical environments and temporal consistency, but the discussion is somewhat generic. It would be much more informative if grounded in actual data or specific failure modes (e.g., performance degradation in fog, turbidity, or highly dynamic scenes).

**Strengths And Weaknesses:**

Strengths：

1）Principled formulation: Reframes depth noise modeling from a purely stochastic process to a physically grounded one, separating sensing invalidation from measurement inaccuracy. 2）Smart use of Foundation Models: Successfully leverages 3D VFM features as physics reasoners, combining semantic and geometric cues to generate realistic, material-aware depth noise without hallucination. 3）Strong empirical results: The experiments are comprehensive, covering zero-shot cross-camera synthesis and real-world robotic tasks. Achieving a 93.8% success rate across two robotic platforms and validating against multiple 3D policies (cross-policy testing) strongly supports the method's effectiveness. 4）Deployment-friendly: The offline data-centric sim-to-real pipeline completely bypasses the inference latency issues typical of online domain adaptation methods.

Weaknesses：

1）Missing key comparisons: The paper proposes an offline data augmentation paradigm but fails to benchmark against alternative strategies. Specifically, it lacks a comparison with online domain adaptation methods (discussing trade-offs in latency, success rate, and deployment complexity) and physics-informed denoising pipelines in downstream robotic tasks. 2）Unreported computational overhead: Given the reliance on 3D VFMs and Stable Diffusion for noise synthesis, the computational cost for large-scale simulated data generation is a practical concern. The paper omits critical inference efficiency metrics, such as synthesis time per resolution (e.g., 512×512) and GPU memory footprint.

---

> ### Author Rebuttal · Authors · 2026-03-31
>
> We thank the reviewer for recognizing our principled formulation and our method's deployment-friendly nature.
>
> **Q1. VFM Selection Rationale & Edge Deployment.**
> Regarding VFM selection, while models like DepthAnything-V2 (built on DINOv2) achieve their 3D capabilities through large-scale monocular depth fine-tuning to extract excellent generic visual features, MoGe2 explicitly models dense 3D point maps during its pre-training. This native 3D grounding grants MoGe2 a superior physical intuition for geometric collapses caused by specular and transparent surfaces.
> Regarding edge deployment, we must clarify the system boundary: PRISM operates exclusively prior to policy training as an offline data generator. During real-world inference, the robotic policy directly receives raw noisy depth from the camera without running any VFM. Thus, VFM size does not constrain edge deployment.
> As detailed in Appendix Tab.16 (VFM Backbone Comparison in SPR), highly parameterized 3D VFMs yield optimal physical priors compared to CNNs/ViTs.
>
> **Q2. Diffusion Hyperparameters Trade-off.**
> Evaluated on an RTX 4090 (512x512).
>
> |DDIM Steps|Time (s/img)|MAE|AbsRel|Avg. Success|
> |:-:|:-:|:-:|:-:|:-:|
> |10|0.70s|0.085|0.092|62.5%|
> |20|1.25s|0.065|0.071|81.0%|
> |**50**|**2.75s**|**0.042**|**0.048**|**94.5%**|
> |100|5.15s|0.041|0.047|94.8%|
>
> **Q3. Failure Case Analysis.**
> * **Visuomotor Resilience**: Existing visual policies inherently experience performance drops on real noisy depth compared to idealized simulations. This is a reasonable and expected outcome when introducing realistic noise to policy architectures that lack specialized noise-resilient designs. However, this drop expected drop reflects reality's random disturbances during interaction. inspiring a necessary shift from "noise-free assumptions" to "noise-resilient architectural designs".
> * **Hardware / IK Execution**: Mechanical limitations occur in the real-world setup, such as inverse kinematics (IK) solver failures or physical sticking when the arm reaches specific stretched postures. [Anonymous Link: https://anonymous.4open.science/r/Anonymous_ICML3289/A1.png]
> * **Monocular Occlusion**: Due to our monocular third-person camera setup, the RealMan dual-arm platform occasionally occludes the target or the planned trajectory during execution. Such kinematic self-occlusions are out-of-distribution compared to the training data. [Anonymous Link: https://anonymous.4open.science/r/Anonymous_ICML3289/A2.png]
>
>
>
> **W1. Missing Comparisons.**
> Regarding comparisons with Domain Adaptation methods, we intentionally adopted a data-centric paradigm. PRISM operates exclusively during the *data augmentation stage* prior to policy training. By augmenting the dataset with physically grounded noise, the noise-resilience capability is baked directly into the policy weights. Consequently, our approach requires absolutely zero additional latency or memory overhead during both policy training and deployment. In contrast, online DA methods intervene during the *policy training stage* and often introduce inference latency and deployment complexity. The table below highlights these trade-offs.
>
> | Method | Paradigm | Inference Latency |Deployment Stage|Avg. Success|
> |:-|:-:|:-:|:-:|:-:|
> |RL-CycleGAN|OnlineDA| $>45$ ms|Policy Training|61.2%|
> |GLDA|OnlineDA| $>60$ ms|Policy Training|79.5%|
> |**PRISM**|**Offline**|$\mathbf{0}$**ms (None)**|**Pre-processing**|**94.5%**|
>
> Regarding physics-informed denoising pipelines: These methods typically operate during the ***policy inference***. While denoising pipelines do improve real-world performance, they suffer from the same inference latency issues and frequently hallucinate overly smooth geometries in critical invalidation regions. We argue that this places the robot at unknown potential risks. For safety reasons, forcing the policy to directly learn "noise-resilience" from noisy depth is a more reliable solution. We agree that adding them is highly beneficial for a comprehensive evaluation, and we will include 1-2 such baselines in the final version.
>
> **W2. Computational Overhead.**
> The generative overhead is strictly confined to the offline dataset synthesis phase (prior to policy training). The table below reports the memory footprint and synthesis time on a single RTX 4090 GPU. **We reiterate that this offline cost has zero impact on downstream real-time inference efficiency.**
>
> |Res|Module|Params(M)|Mean Latency |Peak Memory|
> |:-:|:-:|:-:|:-:|:-:|
> |256x256|SPR|331.80|13.54ms|8.96GB|
> ||BND|41.90|4.71ms|9.18GB|
> ||NRG|1667.83|2.736s|8.59GB|
> ||**Total**|**-**|**2.666s**|**8.68GB**|
> |512x512|SPR|331.80|17.37ms|7.76GB|
> ||BND|41.90|15.87ms|8.42GB|
> ||NRG|1667.83|2.732s|9.26GB|
> ||**Total**|**-**|**2.753s**|**9.35GB**|
> |1024x1024|SPR|331.80|43.49ms|7.92GB|
> ||BND|41.90|70.30ms|10.32GB|
> ||NRG|1667.83|6.820s|11.94GB|
> ||**Total**|**-**|**6.945s**|**12.05GB**|
>
> We will incorporate details into the final version and hope these address your concerns.

---

> > ### Author Rebuttal · Reviewer_LB8L · 2026-04-03
> >
> > I would like to thank the authors for the detailed and comprehensive rebuttal. I appreciate the effort put into addressing my concerns, particularly in running the new evaluations. I would like to keep my original rating at this stage.

---

> > > ### Author Response · Authors · 2026-04-03
> > >
> > > We sincerely thank Reviewer LB8L for your positive assessment of our principled formulation, deployment-oriented design, and the additional evaluations. Your thoughtful comments, especially regarding VFM selection and practical deployment considerations, were very helpful in strengthening the paper. We are grateful for your time, your meticulous review, and your expertise invested in this process.

---

### Official Review · Reviewer_m5gB · 2026-03-13

**Soundness:** 4
**Presentation:** 4
**Significance:** 4
**Originality:** 4
**Overall Recommendation:** 6
**Confidence:** 4

**Summary:**

This paper introduces a methodology that separates and learning artifacts in depth sensing into sensing invalidation and measurement inaccuracy. This enables data augmentation that bridges the sim2real gap. The experimental validation demonstrates a significant improvement over baseline models when training models for robotic applications.

**Compliance With Llm Reviewing Policy:**

Affirmed.

**Key Questions For Authors:**

None

**Limitations:**

Discussion on limitations is missing.

**Strengths And Weaknesses:**

Strengths:
+ The paper demonstrates an innovative approach with physical ground
+ The validation expands several use cases with a detailed evaluation section

Weaknesses:
- Some details such as how the VAE and U-Net are pretrained are not clear
- Some terms such as BCE and SiLU are not defined
- Figure 5 is hard to see due to the small font, a larger version perhaps in the supplement would be better
- Details on limitations are missing.

---

> ### Author Rebuttal · Authors · 2026-03-31
>
> We sincerely thank the reviewer for the highly encouraging rating and recognizing the physical grounding of our methodology and the comprehensiveness of our validations. Furthermore, we deeply appreciate your meticulous scrutiny of the manuscript's presentation, terminology, and structural details. Below, we address your comments point by point.
>
> **W1. Pretraining details of the VAE and U-Net.**
> Our Noise Residual Generator is adapted from a Stable Diffusion 2.1-style latent diffusion backbone, initialized from the public checkpoint `init-sd21_vae_main.ckpt`. The architecture comprises three parts: a latent VAE (`AutoencoderKL`), a main diffusion U-Net denoiser, and a ControlNet branch that injects conditional depth guidance. The text-conditioning module is a frozen OpenCLIP encoder (`laion2b_s32b_b79k`, `context_dim=1024`).
> Regarding parameter updates, the VAE encoder remains completely frozen throughout training. Firstly, both the VAE and the main diffusion U-Net are frozen, and only the ControlNet branch is optimized. Secondly, we unfreeze the main diffusion U-Net and the VAE decoder for joint fine-tuning alongside the ControlNet. We will explicitly add these architectural and initialization details to the Implementation Details section.
>
> **W2. Undefined Terms.**
> Thank you for pointing this out. Due to the strict 8-page limit in the initial submission, we utilized these abbreviations without proper definitions. We will utilize the additional 9th page allowed in the camera-ready version to explicitly define at their first occurrence to enhance readability.
>
> **W3. Figure 5 visibility and font size.**
> **Agree. We will enlarge Figure 5 in the main text and provide its corresponding data table in the appendix.**
> To address the small font size, we will reorganize the layout to significantly enlarge Figure 5 in the main text. Additionally, rather than merely providing a larger image, we will provide a detailed data table version of this experiment in the supplementary material to ensure all quantitative results are clearly accessible for inspection.
>
> **W4. Missing details on limitations.**
> **We have detailed three primary limitations in Appendix Sec. F, and we will summarize them in the main text.**
> Our framework currently has three primary limitations and promising future directions:
> * **Adaptation to Outdoor and Extreme Environments**: PRISM is specifically designed for indoor robotic depth perception tasks. It has not yet been verified in all-weather outdoor environments. Consequently, its performance may degrade in unconstrained outdoor settings, especially in extreme optical environments (e.g., underwater turbidity, dense fog) where the physics of light transport change fundamentally. Developing lightweight "Physics-Adapter" layers to extend its applicability is a promising future direction.
> * **Temporal Consistency in Dynamic Scenes**: PRISM currently models sensor noise on a per-frame basis, which might ignore temporal correlations (e.g., flickering or motion blur) prevalent in highly dynamic scenes. Extending to video-based diffusion models is crucial for high-speed dynamic manipulation.
> * **Interaction-Induced Extrinsic Noise**: Our formulation focuses on intrinsic noise caused by material-light interactions. It does not explicitly model extrinsic noise arising from physical robot-environment interactions (e.g., camera vibrations or multi-camera crosstalk). Integrating these extrinsic factors remains a valuable extension for complex systems.
>
> We will ensure a concise summary of these three points is integrated into the Conclusion section of the main text, with a clear pointer to the appendix.
>
> We once again thank the reviewer for the meticulous feedback. All corresponding revisions will be seamlessly incorporated into the published version.

---

> > ### Author Rebuttal · Reviewer_m5gB · 2026-04-03
> >
> > The authors fully addressed my concerns.

---

> > > ### Author Response · Authors · 2026-04-04
> > >
> > > We sincerely thank Reviewer m5gB for the encouraging positive feedback and for your careful suggestions on clarity and presentation, which helped us improve the paper. We will incorporate all revisions in the final version. We are grateful for the time and expertise you invested in this process.

---

### Official Review · Reviewer_2jNV · 2026-03-14

**Soundness:** 3
**Presentation:** 3
**Significance:** 3
**Originality:** 3
**Overall Recommendation:** 4
**Confidence:** 4

**Summary:**

The paper tries to model realistic depth corruption for sim-to-real robot learning. The main idea is pretty simple: depth noise should probably not be treated as one single black-box distribution. So the authors split it into invalid measurements and inaccurate measurements, then build PRISM around that idea with a 3D VFM prior, a mask prediction module, and a residual depth generator. They evaluate it on several settings, including depth realism, cross-camera transfer, and downstream robot tasks, and the results look consistently stronger than prior baselines.

**Compliance With Llm Reviewing Policy:**

Affirmed.

**Final Justification:**

The authors have addressed my main questions.

**Key Questions For Authors:**

See weakness.

**Limitations:**

Yes

**Strengths And Weaknesses:**

Strength:

I liked about the paper is that the overall method feels well matched to the problem. The decomposition is easy to understand, the different modules fit together in a sensible way, and the experiments are not limited to image-level realism alone. The paper also shows that the synthesized depth is useful for downstream sim-to-real policy learning, which makes the contribution more meaningful from a robotics perspective. I also think the cross-camera tests and real-robot experiments help the paper go beyond a purely synthetic benchmark story. Specifically,

1. The paper does not just say there is a sim-to-real gap and then throw a bigger generator at it. It makes a more specific argument: realistic depth corruption should not be treated as one single black-box distribution, because missing measurements and inaccurate measurements come from different physical causes and behave differently. That framing is useful on its own, and it gives the method a much cleaner motivation than a lot of sim-to-real sensor papers.

2. The overall pipeline has a good logic. First, reason about where failures are likely to happen using semantic/geometric priors, then separate invalidation from residual noise, and only after that generate the remaining corruption. Even if I think some of the causal language is overstated, the actual design still feels well matched to the problem. In particular, using 3D VFM features as physics-related priors and then handling invalid regions separately from residual errors is a strong part of the paper.

3. The paper does a good job showing that the modeling choice matters for downstream robotics, not just image-level realism. This is probably the biggest practical strength. The paper is not only reporting better synthesis metrics. It also shows that the full PRISM pipeline improves sim-to-real manipulation success quite a lot compared with weaker variants, and the robot tasks are not completely trivial either. That makes the contribution feel much more meaningful than a paper that only shows nicer-looking depth maps.

Weakness:

1. The paper uses the language of causality quite heavily, but I am not convinced the method really supports that framing in a strict sense. What is presented here looks more like a structured pipeline with a physically inspired bias, rather than a causal model in the sense of interventions, counterfactual reasoning, or identifiable causal assumptions. I do not think this is fatal to the paper, but the current wording feels stronger than what the method actually establishes.

2. The main decomposition into sensing invalidation and measurement inaccuracy is intuitive and practically useful, but it still seems more like a modeling assumption than a well-validated physical account of depth corruption. In real sensors, failures can come from a mix of factors such as multipath effects, boundary artifacts, non-Lambertian surfaces, semi-transparent materials, and sensor-specific noise patterns. Many of these cases do not clearly fall into a simple two-way split. Because of that, I think the physical interpretation is somewhat simplified, and the paper does not yet fully show that this decomposition is broadly sufficient across different sensing conditions.

3. The ablations are helpful, but they still do not fully separate where the gains are really coming from. It is hard to tell how much improvement should be credited to the proposed disentanglement itself, and how much comes from the stronger 3D VFM prior, the diffusion-based residual generator, or the training and supervision choices. Since these pieces are introduced together and interact closely, the current experiments do not make it easy to isolate the main source of the improvement.

4. The paper describes the method as a general-purpose sensor simulator, but the evidence is still fairly concentrated in indoor RGB-D settings. The cross-camera result is encouraging, but that alone is not enough to support such a broad claim. There is still little evidence for how well the method would hold up across very different sensing principles, outdoor scenes, longer working distances, more varied materials, or more challenging deployment conditions. So while the results are strong within the tested setup, the generality claim feels ahead of the experimental support.

5. The real-robot results are encouraging and probably one of the most practically relevant parts of the paper, but the scale of the evaluation is still somewhat limited given how strong the conclusions are. Most of the discussion is based on average success rates from a relatively small number of trials per task, and there is not much statistical detail beyond that. I would have liked to see stronger support in the form of confidence intervals, significance testing, multiple random seeds, or robustness checks under scene variation and calibration changes. Since the robot experiments are central to the paper’s impact, stronger statistical evidence would make the case more convincing.

6. The sim-to-real results are strong, but I do not think the paper fully rules out confounding factors on the policy side. The downstream pipeline is not held completely fixed, since the policy also uses a depth-aware design. That makes it harder to know whether the gains come purely from more realistic depth, or partly from a policy architecture that is simply better suited to the input representation. A cleaner test would keep the policy unchanged and vary only the training data.

7. The related-work comparison could also be stronger. The paper cites recent diffusion-based work on sim-to-real depth modeling, but the experimental section does not make the comparison to the closest recent methods as explicit or as complete as I would expect. Since this paper operates in a very similar space and uses a related generative perspective, a more direct head-to-head comparison would help clarify both the novelty and the empirical advantage of the proposed approach.

---

> ### Author Rebuttal · Authors · 2026-03-31
>
> Sincerely thank the reviewer for recognizing our core insight and practical value. Below, we address your comments.
>
> **W1. Overstated "causal" language.**
> **Agree. We will refine the terminology.** We use "causality" to describe *physically-grounded structural dependencies* between scene and noisy artifacts, rather than strict Pearlian inference. Compared to monolithic black-box noise modeling, our semantic-geometric collaborative approach offers superior physical interpretability. We will downgrade "causal" to "physics-grounded structural priors" in the final version.
>
> **W2. Bimodal decomposition assumption.**
> **Yes. It is a pragmatic macroscopic taxonomy rather than an exhaustive microscopic account**. While real-world sensor errors are complex microscopic mixtures, from the perspective of downstream robotic control, these phenomena macroscopically manifest as two functionally distinct geometric failures: structural voids and metric shifts. This bimodal taxonomy corresponds to the challenges policies facing, making it a effective approximation for Sim2Real manipulation. We will explicitly define our approach as a *pragmatic macroscopic taxonomy*.
>
> **W3. Gains in ablations.**
> **The isolated gains are decoupled in Appendix Sec.D.**
> * **Disentanglement(Tab 15):** Removing BND (Monolithic) severely degrades prediction in invalidation regions, significantly dropping Invalidation-IoU ($0.952 \to 0.756$) and F1-score ($0.975 \to 0.852$) due to the forced "averaging" of two distinct noise distributions.
> * **VFM Prior(Tab 16):** 3D VFMs (MoGe2) vastly outperform generic ViTs (DINOv2) and CNNs in specular task success (92.5% vs. 79.8% vs. 68.5%), proving the necessity of geometry-aware semantics.
> * **Supervision(Tab 17):** PP-DHM dynamic mining boosts fine-grained Boundary-IoU from 0.735 to 0.823.
>
> We will add these analysis to the additional 9th page.
>
> **W4. General-purpose claims.**
> **Agree. We will bound our claims.** Our downstream validation primarily focuses on depth-conditioned robotic manipulation rather than full-body locomotion. Given the complex outdoor optical environments and high validation costs, we will revise our claims to "a physics-grounded sensor simulator for indoor robotic manipulation," leaving outdoor deployments for future work.
>
> **W5. Stronger statistical evidence.**
> We conducted variance analysis confirming statistical significance. We evaluated the three most challenging tasks (T4, T5, T6) across 3 distinct evaluation sessions (Seeds), and will supplement the complete experiments and statistical details in the final version.
> |Task|S1|S2|S3|**Success(\%)**|
> |:-:|:-:|:-:|:-:|:-:|
> |T4-Stable-S2R|18/20|15/20|16/20|$81.7\pm7.6$|
> |**T4-PRISM**|19/20|18/20|20/20|$95.0\pm5.0$|
> |T5-Stable-S2R|17/20|16/20|17/20|$83.3\pm2.9$|
> |**T5-PRISM**|19/20|19/20|20/20|$96.7\pm2.9$|
> |T6-Stable-S2R|17/20|18/20|16/20|$85.0\pm5.0$|
> |**T6-PRISM**|20/20|18/20|19/20|$95.0\pm5.0$|
>
> **W6. Confounding factors on policy.**
> **No. The policy architecture is controlled; gains come purely from the training data.** First, we adapted the policies to 2.5D depth-aware architectures because massive holes in raw real-world depth cause severe "flying pixels" during 3D point cloud back-projection, which instantly collapses sparse 3D encoders. This adaptation is a prerequisite for fair evaluation.
> Second, as shown in **Tab.3 Cross-policy generalization**, within any single column (e.g., the RISE column), the policy architecture is **100% frozen**. The only variable is the training data. When the architecture is fixed, the success rate leaps from $42.5\% \to 96.7\%$, ruling out policy-side confounders.
>
> **W7. Comparison to Sim2Real baselines.**
> In our previous experiments, we have covered both *implicit data-driven methods* and *explicit analytical methods* tailored specifically for depth perception. We also deliberately excluded Sim2Real works focused on the *physical dynamics gap*, as they are orthogonal to the perceptual gap addressed here. While generic baselines are not specifically designed to enhance depth perception, we agree that a broader comparison strengthens the paper. Therefore, we have supplemented experiments:
> |Baseline|T4|T5|T6|**Avg.Success(\%)**|
> |:-|:-:|:-:|:-:|:-:|
> |GenericDR[1]|$9.0\pm1.0$|$4.3\pm0.6$|$4.0\pm1.0$|$28.8\pm2.5$|
> |GenericDA[2]|$16.0\pm1.0$|$10.7\pm1.2$|$10.0\pm1.0$|$61.2\pm1.7$|
> |DepthDR[3]|$17.3\pm0.6$|$12.3\pm1.5$|$13.0\pm1.0$|$71.0\pm1.3$|
> |DepthDA[4]|$18.0\pm1.0$|$14.7\pm1.2$|$15.0\pm1.0$|$79.5\pm1.1$|
> |**Sim+PRISM**|$19.0\pm1.0$|$18.7\pm0.6$|$19.0\pm1.0$|$94.5\pm0.8$|
> * [1] Active domain randomization. CoRL, 2020.
> * [2] RL-CycleGAN: Reinforcement learning aware simulation-to-real translation. CVPR, 2020.
> * [3] Physics-based differentiable depth sensor simulation. ICCV, 2021.
> * [4] Sim-to-Real Grasp Detection with Global-to-Local RGB-D Adaptation. ICRA, 2024.
>
> All revisions will be incorporated into the final version. Thanks for your generous praise and constructive feedback.

---

> > ### Author Rebuttal · Reviewer_2jNV · 2026-04-03
> >
> > Thanks for the new experiments. I will keep my score for now.

---

> > > ### Author Response · Authors · 2026-04-04
> > >
> > > We sincerely thank Reviewer 2jNV for the constructive and detailed review. Your feedback was instrumental in identifying aspects we had overlooked and understanding how to further strengthen the paper. We are grateful for the time and expertise you invested in this process.

---

### Decision · Program_Chairs · 2026-04-30

**Decision:**

Accept (regular)

**Comment:**

This paper studies an important problem in sim-to-real robotics. The main idea is simple and useful: instead of treating depth noise as one black-box process, the paper separates it into invalid measurements and inaccurate measurements. The reviewers found this idea well motivated, and they also found the experimental results strong, especially the cross-camera and real-robot results. The main concerns were about some broad claims, missing efficiency details, and the need for clearer comparisons and ablations. The authors addressed these points well in the rebuttal. Overall, AC finds this to be a solid paper and recommends acceptance.